# Magmatic surge requires two-stage model for the Laramide orogeny

**Joshua J. Schwartz** [1] ✉, **Jade Star Lackey** [2], **Elena A. Miranda**[1], **Keith A. Klepeis** [3], **Gabriela Mora-Klepeis** [3], **Francine Robles**[1] & **Jonathan D. Bixler** [1]

The Laramide orogeny is a pivotal time in the geological development of western North America, but its driving mechanism is controversial. Most prominent models suggest this event was caused by the collision of an oceanic plateau with the Southern California Batholith (SCB) which caused the angle of subduction beneath the continent to shallow and led to shut-down of the arc. Here, we use over 280 zircon and titanite Pb/U ages from the SCB to establish the timing and duration of magmatism, metamorphism and deformation. We show that magmatism was surging in the SCB from 90 to 70 Ma, the lower crust was hot, and cooling occurred after 75 Ma. These data contradict plateau underthrusting and flat-slab subduction as the driving mechanism for early Laramide deformation. We propose that the Laramide orogeny is a two-stage event consisting of: 1) an arc 'flare-up' phase in the SCB from 90-75 Ma; and 2) a widespread mountain building phase in the Laramide foreland belt from 75-50 Ma that is linked to subduction of an oceanic plateau.

One of the most important periods in the development of western North America occurred at ca. 90-80 million years (m.y.) ago when a series of deeply rooted thrust faults began to uplift and imbricate slices of continental lithosphere located hundreds to thousands of kilometers inland from the coast (Fig. 1a)[1–5]. This major thick-skinned tectonic event, called the Laramide orogeny, lasted ~40–60 m.y., and resulted in mountain building, the formation of foreland basins, and the development of ore mineralization from Canada to northern Mexico and as far east as the Black Hills of South Dakota[1–3]. Nevertheless, despite its widespread impact on the tectonic development of western North America, the exact mechanisms that initiated this major tectonic event remain controversial[4–8].

Most widely cited models argue that the cause of the Laramide orogeny was the flat-slab subduction of a thick oceanic plateau beneath the Southern California Batholith (present-day Transverse Ranges in Southern California) at 90-80 Ma, resulting in shutdown of arc magmatism and cooling of the upper-plate crust[7,9–15]. These 'amagmatic' models resemble the present-day central Andean orogen (27–33°S) where a flattened subducted oceanic slab is associated with

thick-skin deformation and relative magmatic quiescence in the Sierras Pampeanas[16]. In contrast, other models propose that this tectonic transition was caused by the collision of off-shore arc terranes[17,18], or increased lithospheric coupling[5,19]. Important features of these conflicting models are that they make different predictions about the timing and duration of magmatism, cooling, and uplift during the development of the Laramide orogeny.

To resolve these conflicting models, we examined the Southern California Batholith (SCB) that represents a ~500-km-wide, paleo-arc segment of the Mesozoic California arc where prior studies place the initial collision between the conjugate Shatsky plateau and the western North American margin[12,14]. This arc segment lies between the southern Sierra Nevada batholith (SNB) and northern Peninsular Ranges Batholith (PRB) and is now represented by fault-bounded structural blocks that make up the central and eastern Transverse Ranges and the western Mojave province. This region experienced varying degrees of faulting and rigid block rotation related to the development of the San Andreas transform plate boundary[20,21]. We focus on the frontal arc of the Southern California Batholith (SCB) in the Transverse Ranges

[1]Department of Geological Sciences, California State University Northridge, Northridge, CA 91330, USA. [2]Geology Department, Pomona College, Claremont, CA 91711, USA. [3]Department of Geography & Geosciences, The University of Vermont, Burlington, VT 05405, USA. ✉ e-mail: joshua.schwartz@csun.edu

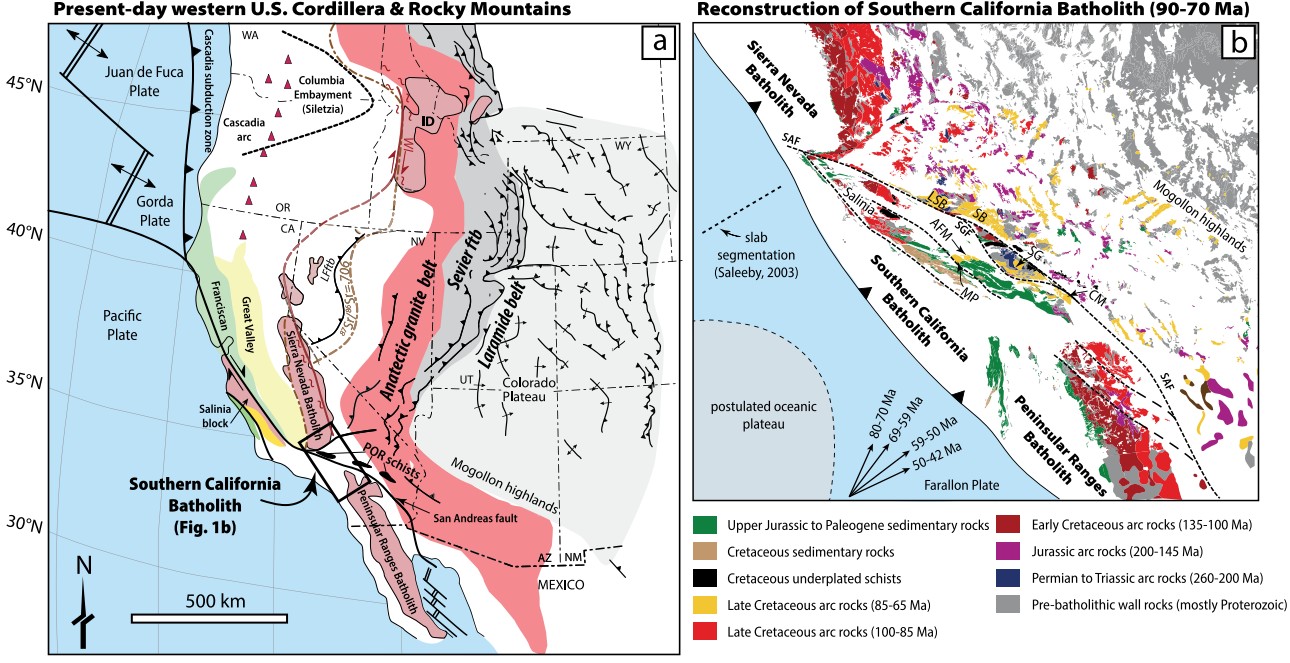

**Fig. 1 | Generalized maps of the US sector of the western North American Cordillera and palinspastic reconstruction of the Southern California Batholith. a** Map of western North American Cordillera showing the current distribution of the Mesozoic and present-day magmatic arc, Sevier fold-thrust belt, Laramide foreland belt, and hinterland (after[42]). The distribution of the anatectic granite belt is after Chapman et al.[32]. **b** Late Cretaceous (ca. 90–70 Ma) tectonic reconstruction of the Cordilleran arc in Southern California (after[48]). The Southern California Batholith (SCB) lies between the northern Peninsular Ranges Batholith and southern Sierra Nevada Batholith. This study focuses on the Late Cretaceous (90–70 Ma) plutonic flare-up in the SCB (yellow rocks), particularly those in the coastal arc in the Transverse Ranges and their relationship to flat-slab subduction models. Our data come from all major structural blocks in the Transverse Ranges (see abbreviations on map). SAF San Andreas fault, SGF San Gabriel fault, LSB Little San Bernardino Mountains, SB San Bernardino Mountains, MP Pine Mountain block, SG San Gabriel Mountains, CM Cucamonga block, AFM Alamo-Frasier Mountain block.

because of its unique location as the proposed site of initial collision[12,14]. As such, the SCB is the key location to test and to resolve conflicting models for the structural, magmatic, and metamorphic response to tectonic changes that affected continental lithosphere in western North America in the Late Mesozoic.

## Discussion and results

### Zircon geochronology

To test models for the Laramide orogeny, we dated 79 zircon- and titanite-bearing samples and compiled >200 Pb/U ages from >4000 km$^2$ in the Southern California Batholith to establish the timing and duration of magmatism, metamorphism and deformation. Areal addition rates were calculated from integrating igneous ages with pluton areas determined from digitized geologic maps of Southern California. Our compilation and new data encompass all major blocks in the SCB including the San Gabriel, Pine Mountain, Alamo-Frazier Mountain, Little San Bernardino, San Bernardino, and Salinian blocks (Fig. 1b and Supplementary file). Igneous zircon ages from calc-alkaline diorites, tonalites, granodiorites and granites reveal 3 discrete pulses of Late Paleozoic to Mesozoic magmatism at 260–210 Ma, 160–140 Ma, and 90–70 Ma (Fig. 2a). These Phanerozoic magmatic events intruded into pre-batholithic Proterozoic basement gneisses that range in age from 1950 to 1100 Ma. The Late Cretaceous pulse culminated in an arc flare-up event which peaked at 85–75 Ma and was associated with widespread, voluminous plutonism throughout the SCB (Figs. 1b and 2b). Although the cause of the 85–75 Ma flare-up in the SCB remains unclear, magmatism persisted in the frontal arc of the SCB until ca. 70 Ma (Fig. 2c and Supplementary file). The Late Cretaceous flare-up in the SCB also temporally overlaps with the eastward migration of Late Cretaceous plutonism into the upper-plate zone of the adjacent PRB (85-74 Ma[22,23]) as well as in northwestern Mexico (100-

45 Ma[24–29]). While magmatism ceased in the SNB by 85 Ma, arc magmatism continued south of the proto-Garlock fault in the SCB and southward into Mexico in the Late Cretaceous reflecting segmentation of the Mesozoic continental arc south of the SNB[12].

Compositionally, the Late Cretaceous plutons in the SCB have typical Cordilleran-arc affinity, which supports their derivation in a continental arc setting. Most plutons are magnesian, metaluminous to weakly peraluminous, and calc-alkalic with strong crustal affinities that reflect mixtures of Proterozoic crust with juvenile mantle sources (Fig. 3a–c)[30,31]. They overlap in composition with arc-related magmas in the SNB, but are generally older and geochemically distinct from Late Cretaceous to Eocene peraluminous leucogranites that make up the two-mica or Cordilleran Anatectic Belt in the hinterland of the western North American Cordillera (Figs. 1 and 3d)[32]. The latter are interpreted to reflect water-absent muscovite dehydration melting and/or water-deficient melting of metasedimentary rocks, and are likely related to the melting of thickened continental crust[32].

An analysis of metamorphic zircon and titanite in the SCB shows that the batholith records high temperatures during the Late Cretaceous transition to thick-skinned, Laramide shortening. The data demonstrate that the arc-flare up event in the SCB was also coeval with high-temperature metamorphism at garnet-granulite to upper amphibolite-facies metamorphism and partial melting in the lower crust of the arc. Evidence for high-temperature metamorphism is preserved in the Cucamonga terrane (eastern San Gabriel Mountains) where metamorphic zircons in gneisses, migmatites and calc-silicates give dates ranging from 86 to 76 Ma at 9-7 kbars[33,34] (Supplementary file). Ti-in-zircon analyses on metamorphic rims indicate temperatures of 800 to 713 °C and garnet-quartz oxygen isotope thermometry yields similar metamorphic temperatures of 835 to 777 °C. These data are consistent with mineral exchange thermometry that give temperatures

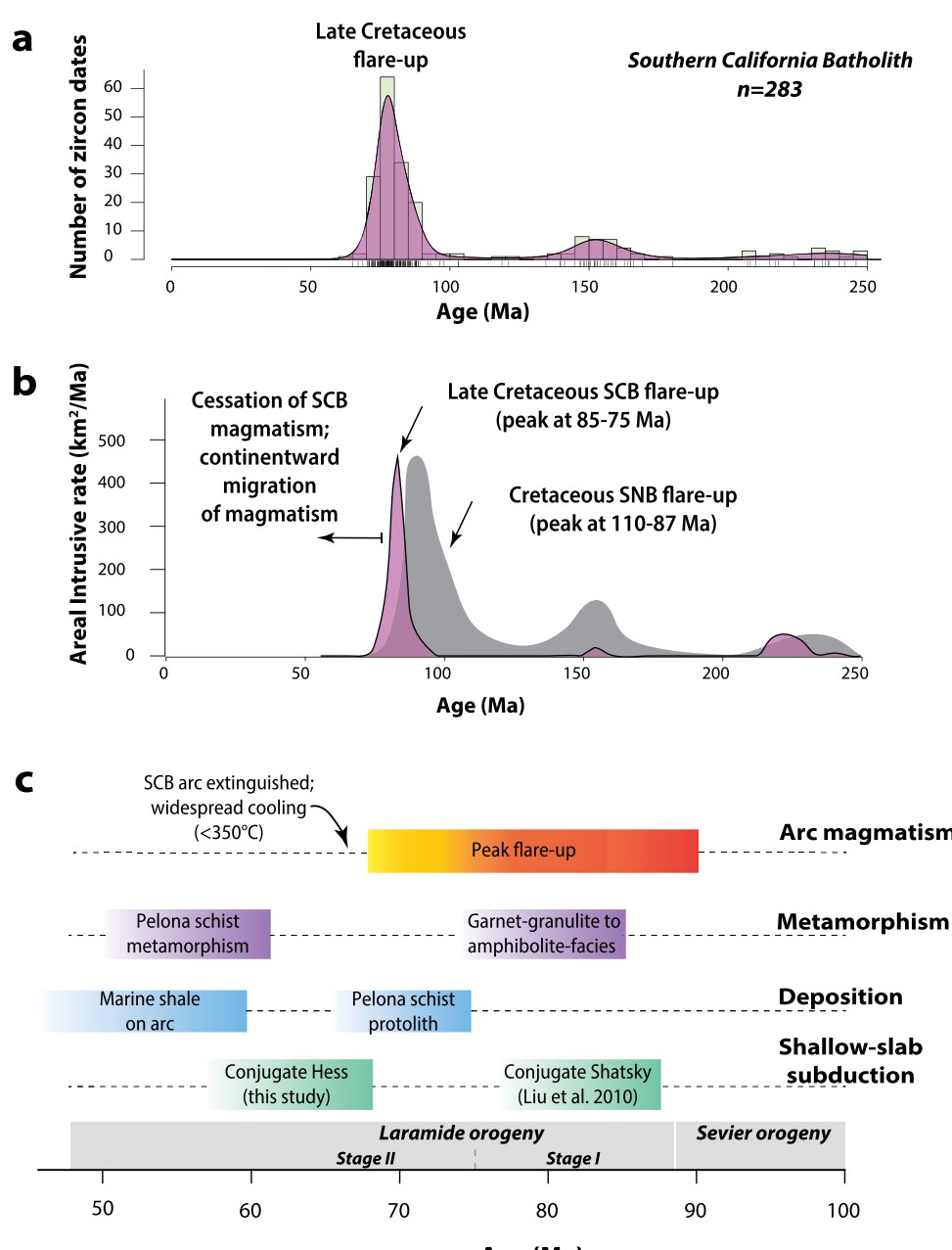

**Fig. 2 | Timing of Mesozoic events in the Southern California Batholith (SCB).** **a** Histogram and kernel density estimates for plutons in the SCB. **b** Calculated areal addition rates (km²/Ma) versus time for the Mesozoic SCB in 5 m.y. bins. Areal addition rates for the Sierra Nevada Batholith are shown for comparison (not to scale)[38]. **c** Temporal evolution of magmatism deformation, metamorphism, deposition, and flat-slab subduction in the SCB. Our new model is compatible with underthrusting of the conjugate Hess oceanic plateau after 75-70 Ma, but is inconsistent with earlier amagmatic models and collision of the putative conjugate Shatsky Rise at ca. 88 Ma. The two stages of the Laramide orogeny are illustrated at the bottom of C. Stage I involved a magmatic flare-up event associated with granulite-facies metamorphism in the SCB, and basement-involved thrusting and basin formation in SW Montana. Stage II involves rapid cooling of the SCB, widespread basement-involved thrusting, and basin formation in Utah, Colorado and Wyoming. We attribute this latter part of the Laramide orogeny to underthrusting of the conjugate Hess oceanic plateau beneath the SCB. SCB Southern California Batholith, SNB Sierra Nevada Batholith.

of 800-775 °C[34]. In the Coast Ridge Belt (Santa Lucia Mountains, Salinia block), Kidder et al.[35] also report peak metamorphic pressures of 800 °C at 7.5 kbars at 81 to 76 Ma. These results are significant because they show that the root of the arc was hot and partially molten through ca. 76 Ma.

**Implications for Late Cretaceous high-temperature arc processes in the SCB**
Our compilation of igneous zircon dates from the frontal arc of the SCB document a widespread surge of Late Cretaceous magmatism from 90 to 70 Ma. This surge occurred throughout all major structural blocks that make up the ca. 500-km-wide segment of the Late Cretaceous arc system (Figs. 1b and 2). Outside of the SCB, coeval magmatism also occurred in the eastern Peninsular Ranges batholith, the Sonora and Sinaloa batholiths of northern Mexico, as well as in the back-arc of the SCB, which is now represented by dispersed Cretaceous plutons in the Mojave Desert[36]. The widespread extent of the SCB and their arc-like geochemical features (Fig. 3)[31,37] suggests that mechanisms for generating large volumes of melt were still active throughout the Late Cretaceous. Arc flare-ups similar in magnitude

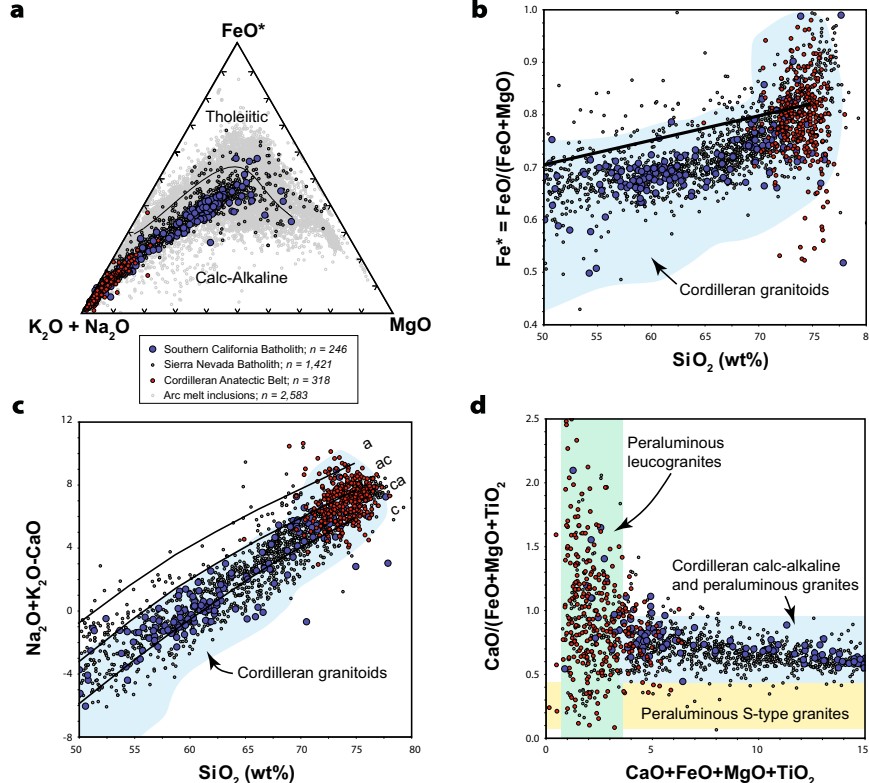

**Fig. 3 | Major-element geochemistry plots for Late Cretaceous arc rocks in the coastal batholiths of the SCB and SNB (blue and black dots) versus per-aluminous leucogranites of the Cordilleran anatectic belt (red dots). a** AFM diagram. **b** Fe* vs. SiO$_2$. **c** Modified alkali lime index vs. SiO$_2$. **d** Major element compositional variation plot. In general, plots show that granitoids from the SCB overlap those of the SNB and arc melt inclusions (gray dots). Peraluminous leucogranites of the Cordilleran Anatectic Belt are geochemically distinct and unrelated to rocks in the coastal batholiths. Melt inclusion data are from the GEOROC database and SNB data are compiled from the NAVDAT database. Fields for Cordilleran granitoids after ref. [87]. FeO* = FeO + (Fe$_2$O$_3$*0.8998). a alkali, ac alkali calcic, ca calc-alkalic, c calcic.

and duration also occurred in adjacent sectors of the arc slightly before the SCB event (e.g., central and southern SNB at 110-87 Ma[38] and northern PRB at 99-91 Ma[39]) and afterward (e.g., Sonora at ca. 71 Ma)[28] (Fig. 2b). Importantly, our data from the SCB contradict existing amagmatic models that invoke underthrusting of the conjugate Shatsky plateau beneath the SCB from 88-75 Ma and removal of the lower crust and lithospheric mantle during the beginning of the Lar-amide orogeny[7,12,14].

Metamorphic ages and thermometry in the Cucamonga and Salinian granulites also demonstrate that the lower crust of the SCB was hot and partially molten through 75 Ma, and these features cannot be explained by existing amagmatic models. The presence of this hot, arc root provides further evidence that high-temperature arc processes were operating until 75-70 Ma and shut-down of the frontal arc did not occur until after 70 Ma. This observation is illu-strated in Fig. 4, which shows a compilation of time-temperature profiles derived from mineral thermochronology from the major structural blocks in the SCB. These data highlight two important features of the SCB: (1) the Late Cretaceous flare-up in the SCB was coeval with high-temperature metamorphism within the arc; and (2) termination of arc magmatism in this region was associated with an abrupt phase of rapid regional cooling of the SCB below 350 °C at ca. 75-70 Ma (Fig. 4). We interpret this regionally extensive and rapid cooling event to record the onset of regional refrigeration of the SCB due to flat-slab subduction involving a cold, oceanic plateau and tectonic underplating of trench sediments beneath the SCB after 75-70 Ma[40].

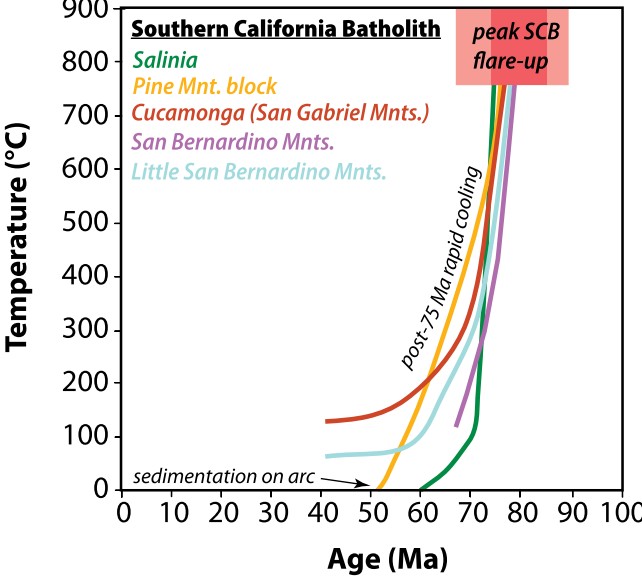

**Fig. 4 | Time-temperature cooling curves for the major structural blocks of the Southern California Batholith.** Data show that the arc flare-up was followed by widespread, post-75 Ma cooling below 350 °C in all structural blocks. These data support models for regional cooling of the SCB after 75-70 Ma due to under-thrusting of the conjugate Hess oceanic plateau[40]. Data compiled from this study and others[33,35,37,78,88].

## Post 75-70 Ma underthrusting of the conjugate Hess Rise oceanic plateau

One of the key results of our work is that flat-slab subduction beneath the SCB post-dates the beginning of Laramide deformation by at least 15 Myr. Consequently, underthrusting of the conjugate Shatsky oceanic plateau cannot be called upon as the driver for thick-skin deformation in the western US prior to ca. 75 Ma. Moreover, our data from the SCB show no evidence for collision or shutdown of magmatism associated with the postulated conjugate Shatsky oceanic plateau at ca. 88 Ma[14]. This observation can be reconciled by recent plate reconstruction models, which show that the Shatsky Rise may never have erupted onto the Farallon oceanic plate[18,41]. If correct, there may not have been a conjugate Shatsky oceanic plateau on the Farallon plate at all.

Although the conjugate Shatsky model fails to explain our data in the SCB, several existing data sets do support a link between Laramide deformation and flat-slab subduction after 75 Ma. For example, the general timing of major thick-skin, basement-cored thrusting, increased exhumation and basin development in Utah, Colorado, and Wyoming occurred from 70-50 Ma[3,42,43], which agrees well with the timing of underthrusting of the conjugate Hess Rise oceanic plateau after 75 Ma[7]. In the SCB, the presence of underplated schists is commonly cited as evidence for flat-slab subduction[44], and our geo- and thermochronological results also support a post-75 Ma emplacement model for the schists. Underplated schists in the Transverse Ranges have zircon age distributions with maximum depositional ages ranging from 75-68 Ma (Pelona Schist[44,45]) and muscovite $^{40}Ar/^{39}Ar$ metamorphic ages from high-pressure/low-temperature mafic and quartzofeldspathic schists are no older than 64 Ma[44,46,47]. Field observations in the Pelona schist also show no evidence for partial melting or intrusion by Cretaceous plutons. These textural and temporal constraints indicate that deposition, metamorphism and underplating of schists beneath the SCB took place after the termination of the SCB flare up event after 75-70 Ma. Therefore, these data are also consistent with flat-slab subduction of the conjugate Hess Rise after 75-70 Ma.

Spatial and temporal trends in sedimentary provenance within California forearc sediments also show a pronounced and sudden influx of continent-derived detritus to the southern California margin at ca. 75 Ma. Jacobson et al.[44] and Sharman et al.[48] argued that this sudden influx reflects the development of a geomorphic breach within the Cretaceous arc and an associated rapid migration of forearc drainages into the continental interior. The timing of this breach is ~15 Myr younger than postulated plateau underthrusting in prior models, but is compatible with our model for the arrival of the conjugate Hess Rise after 75-70 Ma (Fig. 4). These data are also consistent with Late Cretaceous eastward migration of arc magmatism away from the coastal arcs after 75 Ma which has been interpreted to reflect a shallowing of the subduction angle over time[32,42,49]. We illustrate these features in our model (Fig. 5), which shows the tectonomagmatic evolution of the SCB-arc segment from 85-50 Ma.

## A two-stage model for the Laramide orogeny

Data from the SCB provide critical new information that allows us to resolve the debate about the link between upper-plate deformation in the western North American Cordillera and the kinematics and geometry of the down-going plate at the beginning of the Laramide

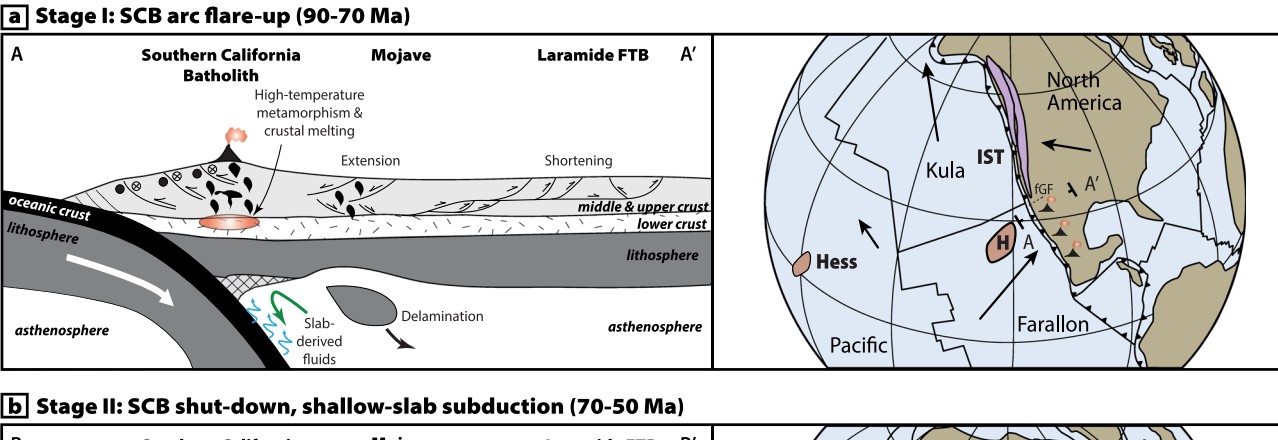

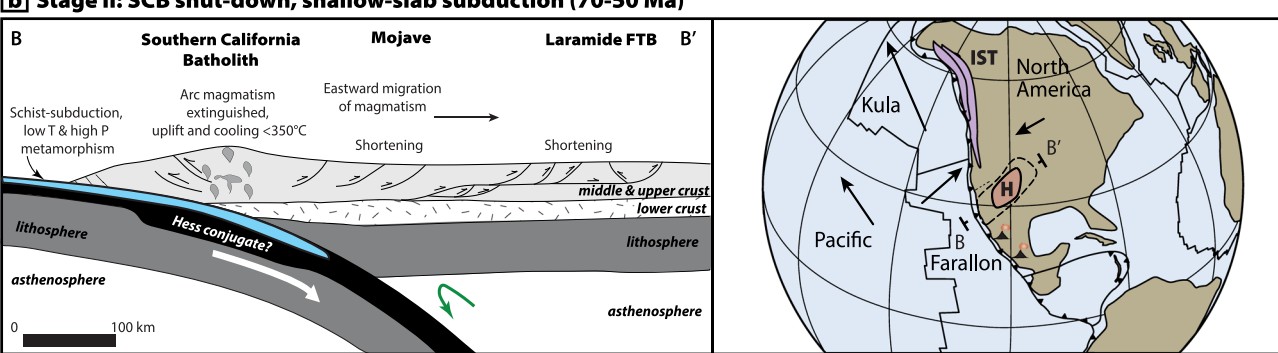

**Fig. 5 | Schematic models for the two-stage evolution of the Southern California Batholith. a** Late Cretaceous arc flare-up in the SCB occurred from 90–70 Ma. Extension in the backarc occurred in the eastern Mojave region and may be related to lithospheric delamination[19], while shortening occurred in the Laramide thrust belt. This phase coincides with possible collision and dextral (northward) translation of the Insular superterrane (IST)[18] that may have occurred north of the SCB and the future Garlock Fault (fGF). **b** Cessation of magmatism, rapid cooling after 75-70 Ma in the SCB, and underthrusting of the Pelona schist is linked to flat-slab subduction and underthrusting of the Hess conjugate (H). Eastward (continentward) migration of the arc is associated with basement-involved thrusting and basin formation in the Laramide fold and thrust-belt. Existing mainstream models focus on this stage of the Laramide orogeny and our data show that this event occurred no earlier than 75-70 Ma and involved the Hess conjugate (rather than the Shatsky conjugate). Global plate reconstructions are modified after[42].

orogeny. Our results show clearly that arc magmatism was robustly active through ca. 70 Ma in the SCB, and consequently, the postulated underthrusting of an oceanic plateau must have occurred after 75-70 Ma. Therefore, flat-slab subduction of the conjugate Shatsky plateau cannot explain the transition from thin- to thick-skin deformation in western North America at ca. 90-80 Ma. This conclusion is supported by sedimentological and thermochronological studies on Mesozoic sediments in southwest Montana, which provide evidence for the early onset of Laramide-style deformation well before 80 Ma[8,50,51]. These data are problematic from the standpoint of an early flat-slab event because the timing of basin formation predates the arrival of the conjugate Shatsky plateau in all flat-slab models[7,12,14]. Moreover, the southwest Montana basins are well outside the commonly cited corridor of Laramide deformation caused by flat-slab underthrusting[15]. These relationships coupled with our data from the SCB lead us to the conclusion that the Laramide orogeny cannot have a single driving mechanism.

Therefore, we propose that the Laramide orogeny is a composite tectonic event consisting of two distinct stages: (1) an early phase at 90 to 75 Ma, which took place during arc flare-up activity in the SCB, and (2) a more-widespread phase of basement-involved thrusting and basin formation in the continental interior of the Laramide foreland belt from 75 to 40 Ma. In this model, the driving mechanism for initial Laramide deformation in southwest Montana and northern Wyoming is not related to flat-slab subduction and requires another cause. One possibility is that deformation resulted from the oblique collision of the Insular superterrane with western North America at ca. 100-85 Ma (the 'hit' phase of the 'hit-and-run' model[17,18]). Data from our study indicate that this collision would have occurred north of the segmented California arc at the SNB-SCB boundary. In support of this idea, we note that the Insular superterrane collision is predicted to have resulted in termination of arc magmatism as observed in the SNB at ca. 85 Ma; however, our data show that Late Cretaceous continental-arc magmatism continued in the SCB and in northwestern Mexico through the Late Cretaceous. In addition, underplated schists beneath the SCB record sedimentation after ca. 70 Ma, which was followed by high- to moderate-pressure/low-temperature metamorphism. Together, these data support models for continued subduction beneath southern California after magmatism had ceased in the SNB. Thus, our observations are potentially compatible with early Laramide deformation resulting from a 'hit-and-run' collision[18]; however, we also note that paleomagnetic data generally suggest more southerly latitudes for the Insular superterrane at this time. Therefore, the initial cause for early Laramide deformation remains enigmatic and a fruitful avenue of future work.

We attribute the second stage of the Laramide orogeny from 75-40 Ma to flat-slab subduction of the conjugate Hess plateau beneath the SCB following the termination of flare-up magmatism in the SCB (Fig. 5). In our model, subduction of the conjugate Hess plateau (rather than the conjugate Shatsky plateau) was the primary driver for uplift and rapid cooling in the SCB, shallow-slab subduction, schist underplating, and basement-involved thrusting and ore mineralization in the Laramide foreland belt from 70-40 Ma. This phase of Laramide deformation is most closely linked to shallow-slab processes and is associated with eastward migration of Laramide magmatism at least through the Late Cretaceous. Collectively, our new data from the SCB demonstrate that multiple driving mechanisms are required to explain the diverse and previously conflicting datasets for the development of Laramide orogeny from 90-40 Ma.

## Methods
### Sample preparation, imaging and analysis
All samples were processed at California State University, Northridge, where they were crushed and pulverized by a jaw crusher and disk mill, respectively, and then run over a Wilfley water table to achieve density separation. The densest outputs of the water table were dried in a 60 °C oven and then sieved to remove grains larger than 250 μ. Grains <250 μ were subjected to a hand-held magnet to remove iron filings and then run through a Frantz isodynamic separator at 0.1, 0.5, 1.0, and 1.5 amps (side tilt = 5°, front tilt = 20°) to remove magnetic minerals. The remaining material was poured into methylene iodide to separate the dense zircon grains from other minerals. Approximately 50–150 zircon grains per sample were placed onto double-sided tape and mounted in epoxy, ground, and polished. The epoxy mounts were imaged at California State University, Northridge using a plane light microscope, as well as a Gatan MiniCL detector with a FEI Quanta 600 SEM, to identify imperfections and spots to target with the laser. Examples of cathodoluminescence (CL) images of zircons and selected spots are reported in Supplementary Fig. 1. A summary of sample locations, dates and primary standards is listed in Supplementary Dataset 1. Zircon isotope ratios are provided in Supplementary Datasets 2 and 3, and zircon trace-element data are provided in Supplementary Dataset 4 and 5. Titanite isotope ratios and trace element data are included in Supplementary Dataset 6.

### SHRIMP-RG U-Pb zircon geochronology
Data were collected at the USGS-Stanford Ion Microprobe Laboratory at Stanford University, California. Two epoxy mounts that contained zircons from the 2012 field season in the Cucamonga block were analyzed during one analytical session in 7–8th September 2012. Zircon geochronology standard R33 (419 Ma quartz diorite zircon[52]) was added to all mounts and MADDER, a Stanford University in-house compositional standard was added to one mount (12CSUN4). The zircons were aligned in $1 \times 6$ mm rows on double-sided tape that was placed on glass slides and then cast in a 25 mm diameter by 4 mm thick epoxy disc. Mounts were ground and polished to a 1 μm finish, washed with a 1 N HCl solution and thoroughly rinsed in distilled water, and dried in a vacuum oven. Mounts were coated with ~100 Å Au layer and were inspected to ensure uniformity and conductivity before loading into the pre-load instrument chamber. The mounts were stored at high pressure (10-7 torr) for several hours before being moved into the source chamber of the SHRIMP-RG to minimize degassing of the epoxy and isobaric hydride interferences and masses 204-208.

Analyses were performed on the SHRIMP-RG ion microprobe at the USGS-Stanford laboratory utilizing an $O_2^-$ primary ion beam, varying in intensity from 4.3 to 6.4 nA, which produces secondary ions from the target that were accelerated at 10 kV. The analytical spot diameter was between ~15 and 20 μ and a depth of ~1–2 μ for each analysis performed in this study. Prior to every analysis, the sample surface was cleaned by rastering the primary beam for 60–120 s, and the primary and secondary beams were auto-tuned to maximize transmission. The duration of this procedure typically required 2.5 min prior to data collection. The acquisition routine included $^{89}Y^+$, 9-REE ($^{139}La^+$, $^{140}Ce^+$, $^{146}Nd^+$, $^{147}Sm^+$, $^{153}Eu^+$, $^{155}Gd^+$, $^{163}Dy^{16}O^+$, $^{166}Er^{16}O^+$, $^{172}Yb^{16}O^+$), a high mass normalizing species ($^{90}Zr_2^{16}O^+$), followed by $^{180}Hf^{16}O^+$, $^{204}Pb^+$, a background measured at 0.045 mass units above the $^{204}Pb^+$ peak, $^{206}Pb^+$, $^{207}Pb^+$, $^{208}Pb^+$, $^{232}Th^+$, $^{238}U^+$, $^{232}Th^{16}O^+$, and $^{238}U^{16}O^+$. Measurements were made at mass resolutions of M/ΔM = 8100–8400 (10% peak height), which eliminated interfering molecular species, particularly for the REE. Analyses consisted of 5 peak-hopping cycles stepped sequentially through the run table. The duration of each measurement ranged between 15 and 25 min on average. Count times for most elements was between 1 and 8 s, with increased count times ranging from 15 to 30 s for $^{204}Pb$, $^{206}Pb$, $^{207}Pb$, and $^{208}Pb$ to improve counting statistics and age precision. R33 was analyzed after every 3–5 unknown zircons. Average count rates of each element were ratioed to the appropriate high mass normalizing species to account for any primary current drift, and the derived ratios for the unknowns were compared to an average of those for the standards to determine concentrations. Spot-to-spot precisions (as measured on the

standards) varied according to elemental ionization efficiency and concentration.

Data reduction for geochronologic results followed the methods described by Williams[53], and Ireland & Williams[54], and used the MS Excel add-in programs Squid 2.51 and Isoplot 3.76 of Ken Ludwig[55,56]. The data were reduced using the Squid 2.51 reduction parameters. The measured $^{206}Pb/^{238}U$ was corrected for common Pb using $^{207}Pb$, whereas $^{207}Pb/^{206}Pb$ was corrected using $^{204}Pb$ following methods by Tera and Wasserburg[57] and Stacey and Kramers[58]. The common Pb correction was based on a model Pb composition from Stacey and Kramers[58]. No addition error was propagated for the uncertainty in the common Pb composition. All reported $^{206}Pb/^{238}U$ and $^{207}Pb/^{206}Pb$ model ages and uncertainties ($2\sigma$) include error summed in quadrature from the external reproducibility ($1\sigma$ SD) of the standard R33 during an individual analytical session (16-24 h). The $1\sigma$ standard error of the mean for the reproducibility of the standard was also propagated into the final calculated $^{206}Pb/^{238}U$ weighted mean age.

Weighted average and Tera-Wasserburg Concordia intercept ages were calculated using SQUID2 processed data in Isoplot 3 add-on for Microsoft Excel[57]. The Concordia and weighted average plots that express the crystallization ages for the samples used MSWD to distinguish overdispersion within a sample. MSWD is defined as:

$$MSWD = f^{-1} \sum (\Delta y_i^2 / \sigma_i^2) \qquad (1)$$

Where $f = (n\text{-}2)$ degrees of freedom, and $n$ represents the total number of data points, $\Delta y_i = y_i - ax_i - b$, is the deviation of the $i$th point and $\sigma_i^2 = \sigma^2 (\Delta y) = a^2 \sigma_{xi} + \sigma_{yi}^2$, is the square of the error. An MSWD close to or equal to 1 occurs if the assigned error is the only cause of the scatter. A value greatly exceeding an MSWD of 1 is due to either: 1) non-analytical errors, such as a geologic phenomenon that creates the deviation from the mean; or 2) an underestimation of the assigned error. MSWD values that are less than 1 are a product of possible overestimation of the analytical error or an unrecognized error correlation.

## SHRIMP-RG trace-element methods

MAD-green (4196 ppm U[59]) was used as trace element standards and precision generally ranged from about ±3% for Hf, ±5-10% for the Y and HREE, ±10–15%, and up to ±40% for La which was present most often at the ppb level (all values at $2\sigma$). Trace elements (Y, Hf, REE) were measured briefly (typically 1 to 3 s/mass) immediately before the geochronology peaks in mass order. All peaks were measured on a single EPT® discrete-dynode electron multiplier operated in pulse counting mode. Analyses were performed using 5 scans (peak-hopping cycles from mass 46 through 254), and counting times on each peak were varied according to the sample age as well as the U and Th concentrations in order to improve counting statistics and age precision.

## LA-SF-ICPMS U-Pb zircon and titanite geochronology (CSUN)

Uranium-lead ratios were collected using a ThermoScientific Element2 SF-ICPMS at California State University Northridge coupled with a Teledyne Cetec Analyte G2 Excimer Laser (operating at a wavelength of 193 nm). Prior to analysis the Element2 was tuned using the NIST 612 glass standard to optimize signal intensity and stability. Laser beam diameter was ~25 μ for zircon and ~40 μ for titanite at 10 Hz and 75-100% power. Ablation was performed in a HelEx II Active 2-Volume Cell™ and sample aerosol was transported with He carrier gas through Teflon-lined tubing, where it was mixed with Ar gas before introduction to the plasma torch. Flow rates for Ar and He gases were as follows: Ar cooling gas (16.0 NL/min); Ar auxiliary gas (1.0 NL/min); He carrier gas (~0.3–0.5 NL/min); and Ar sample gas (1.1-1.3 NL/min). Isotope data were collected in E-scan mode with magnet set at mass 202, and RF Power at 1245 W. Isotopes measured include $^{202}Hg$, $^{204}(Pb+Hg)$, $^{206}Pb$, $^{207}Pb$, $^{208}Pb$, $^{232}Th$, and $^{238}U$. All isotopes were collected in counting mode with the exception of $^{232}Th$ and $^{238}U$ which were collected in

analog mode. Analyses were conducted in a ~40 min. time resolved analysis mode. Each zircon and titanite analysis consisted of a 20-second integration with the laser firing on sample, and a 20 s delay to purge the previous sample and move to the next sample. Approximate depth of the ablation pit was ~20–30 μ.

For zircon, the primary standard, 91,500, was analyzed every 10 analyses to correct for in-run fractionation of Pb/U and Pb isotopes. The secondary zircon standard (Temora-2) was also analyzed every ~10 analyses to assess reproducibility of the data. U-Pb analysis of Temora-2 during all analytical sessions yielded concordant results and error-weighted average ages of 411 ± 0.4 Ma ($n = 280$) which is within 1.4 to 1.8% uncertainty of the accepted ages of 416.8–418.4 Ma[52,60]. The quoted uncertainties in the text, Supplementary Figure 2 and Supplementary Dataset 1 are reported as 2SE internal calculated from Iolite and IsoplotR[61,62]; however, when compared to data from other laboratories, we assign a 2% uncertainty to all dates to account for reproducibility of standards during analyses (Supplementary Dataset 1).

Zircon dates are reported using the $^{206}Pb/^{238}U$ date for analyses <1100 Ma, and the $^{207}Pb/^{206}Pb$ date for those >1100 Ma. For zircons younger than 1100 Ma, the $^{207}Pb/^{206}Pb$ date is an unreliable indicator of discordance due to low abundances of measured $^{207}Pb$. For these zircons, discordance is calculated as the percent difference between the $^{207}Pb/^{235}U$ date and $^{206}Pb/^{238}U$ date. Corrections for minor amounts common Pb in zircon were made on $^{206}Pb/^{238}U$ dates following methods of Tera and Wasserburg[57] using measured $^{207}Pb/^{206}Pb$ and $^{238}U/^{206}Pb$ ratios and an age-appropriate Pb isotopic composition of Stacey and Kramers[58]. Zircons with large common Pb corrections (e.g., analyses interpreted as having ~20% or greater contribution from common Pb) were discarded from further consideration. No corrections were made on $^{207}Pb/^{206}Pb$ dates due to large uncertainties in measured $^{204}Pb$. For Proterozoic-age samples, we report the upper intercept of the $^{207}Pb/^{206}Pb$ dates as the best approximation for the age of the sample.

In samples that exhibited multiple age populations, CL images of samples were examined for textural evidence (Supplementary Fig. 1). In granulitic zircons from the Cucamonga terrane (eastern San Gabriel Mountains), zircons display rounded shapes and complex zonation patterns often consisting of thin luminescent overgrowths (e.g., Supplementary Fig. 1c). These overgrowths are similar to those observed in other high-grade metamorphic terranes (e.g. refs. 63,64,). Consequently, we interpret them as metamorphic in origin and assign them as such in Supplementary Dataset 8. In cases, where older and younger populations are present, we interpret the older age as the likely protolith crystallization age and the younger grain reflect the timing of metamorphism. In some, but not all instances, chondrite-normalized, rare-earth-element patterns show depletions in heavy-rare-earth-element concentrations that indicate that these zircons grew in the presence of metamorphic garnet (e.g., Supplementary Fig. 3c). We targeted these zircons for Ti-in-zircon petrochronology analysis; however, many rims were too thin (<20 μ) to measure with confidence and so we report only a few analyses for these high-grade samples. Overall, our Ti-in-zircon results match oxygen isotope temperatures and those from prior studies of the Cucamonga terrane[34].

For titanite, the primary age standard, MKED, was analyzed every 10 analyses to correct for in-run fractionation of Pb/U and Pb isotopes. The $^{207}Pb/^{206}Pb$ isotope-dilution thermal-ionization mass spectrometry (ID-TIMS) age of 1521.02 ± 0.55 Ma[65] was chosen as the primary reference titanite for U, Th, and Pb isotopes. To assess accuracy and precision, two secondary reference titanites were analyzed every ~10 analyses for U, Th, and Pb isotopes (BLR (1047.4 ± 1 Ma ID-TIMS age[66]) and Fish Canyon (28.4 ± 0.05 Ma $^{206}Pb\text{-}^{238}U$ ID-TIMS age[67]). Depending on abundance, between 30-50 titanite grains per sample were picked from polished thin sections and analyzed. Data were reduced with Iolite[61,68]. Titanite data presented in this study are corrected for

common Pb using a regression on the Tera-Wasserburg Pb/U isochron, where the y-intercept will yield the $^{207}Pb/^{206}Pb$ isotopic ratio of the common Pb for each individual sample. Titanite ages are presented as lower intercepts calculated from regression of $^{207}Pb/^{206}Pb$ and $^{238}U/^{206}Pb$ data by IsoplotR[62]. The quoted titanite dates in the text and tables are internal 2SE, and when compared to data from other laboratories we assign 2% uncertainties based on reproducibility of standards during analyses. In this study, inverse isochron ages for secondary standards using age appropriate initial Pb ratios from Stacey and Kramers[58] were $1063 \pm 3$ Ma for BLR (MSWD 6, initial $^{207}Pb/^{206}Pb$ 0.91) and $27.2 \pm 2.5$ Ma (MSWD = 2.5, initial $^{207}Pb/^{206}Pb$ 0.83) for Fish Canyon tuff titanites. The age for Fish Canyon overlaps within analytical uncertainty and the age of BLR is within 1.5% of the accepted value. The $^{207}Pb$-corrected $^{206}Pb$-$^{238}U$ dates for the unknown titanite were calculated using a free regression and data are shown in Supplementary Figure 4. All data including standards are reported in Supplementary Dataset 6 and are in summarized in Supplementary Dataset 1.

### Zircon and titanite LA-SF-ICPMS trace element geochemistry

Trace elements were measured simultaneously with U-Pb isotopes by LA-SF-ICPMS as described above using Zr and Ca as internal standards for zircon and titanite respectively. For zircon we use nominal values of 43.14 % Zr, and for titanite we use 19.2 wt.% Ca. Trace element data were reduced using Iolite[61,68] and concentrations calculated relative to NIST-612 as a primary standard. BHVO-2G was analyzed as a secondary standard to assess reproducibility of the data. For zircon, model Ti-in-zircon temperatures were calculated using the Ferry and Watson[69] calibration. All samples contain quartz fixing the $aSiO_2$ at unity. Samples from the Cucamonga granulites are associated with granulite-facies mineral assemblages containing rutile. For these samples we estimate the activity of $TiO_2$ at unity, and for samples that lack rutile we assume a value of 0.6 based on the presence of ilmenite.

For titanite, model Zr-in-titanite temperatures were calculated following the methods from Hayden et al.[70]. Temperature uncertainties result from analytical uncertainties in the Zr measurements in titanite, which were <10% ($2\sigma$). We assume an activity of $TiO_2 = 0.7$ based on the scare presence of rutile or ilmenite in analyzed samples, and activity of $SiO_2$ was assumed to equal 1.0 due to the presence of quartz. Pressures were estimated at 0.6 GPa for mid-crustal samples[31]. Given uncertainties in measurements, pressure and activities, we conservatively apply an uncertainty of $\pm50$ °C to all our temperature estimates.

### Oxygen isotope thermometry

We report bulk-mineral oxygen isotope results from 6 samples in the Cucamonga terrane to establish temperatures of metamorphism during granulite-facies metamorphism. Thermometry is based on quartz-almandine/grossular fractionation factors described in Valley et al.[71]. Oxygen isotope ratios ($\delta^{18}O$ in ‰ Vienna Standard Mean Ocean Water (VSMOW)) were measured on 1.5–2.5 aliquots of hand-picked garnet and quartz liberated from 1 cm cores drilled around garnet crystals identified in cut quartzite slabs. The garnet and surrounding quartz were liberated by standard crushing, gravimetric, and magnetic susceptibility techniques. Concentrates were washed in concentrated (37 molar) hydrochloric acid. Oxygen isotopes were analyzed at the University of Oregon Stable Isotope Laboratory by laser fluorination as described in Lackey et al.[72]. Exchange temperatures have uncertainty (2 S.D.) of $\pm 25$ °C based on propagated uncertainty $\delta^{18}O$ on replicate analyses of each garnet and quartz pair of 0.1 to 0.2‰. Data are provided in Supplementary Dataset 7.

### Bulk rock geochemistry

Bulk-rock samples were powdered in an alumina ceramic shatter-box. Powders were mixed with a 2:1 ratio of SpectroMelt A10 lithium tetra

borate flux and melted at 1000 °C for ~20 min to create glass beads at California State University, Northridge. Beads were repowdered, refused following the initial melting parameters, and polished to remove carbon from the flat bottom where analysis occurs. Following procedures outlined in Lackey et al.[72], glass beads were analyzed at Pomona College for major ($SiO_2$, $TiO_2$, $Al_2O_3$, $Fe_2O_3$, MnO, MgO, CaO, $Na_2O$, $K_2O$, $P_2O_5$) and trace (Rb, Sr, Ba, Zr, Y, Nb, Cs, Sc, V, Cr, Ni, Cu, Zn, Ga, La, Ce, Pr, Nd, Hf, Pb, Th, U) elements by X-ray fluorescence (XRF). Beads were analyzed with a 3.0 kW Panalytical Axios wavelength-dispersive XRF spectrometer with PX1, GE, LiF 220, LiF 200, and PE analyzer crystals. New bulk rock data are provided in Supplementary Dataset 9 along with compiled data from the NAVDAT database (https://www.navdat.org/).

### Areal addition rate calculations

We use igneous zircon ages collected in this study with published dates[30,33,36,37,59,73–86], and the duration of individual pluton emplacement to calculate areal addition rates for Mesozoic magmatism in the SCB. The midpoint of each pluton emplacement event was then binned within 5-million-year time increments. Areal intrusive rates ($km^2/Ma$) were calculated by dividing the area of each pluton (obtained from geologic maps) by the temporal span of emplacement. The final graph produced by this process plots magmatic addition rate vs. bin midpoint and provides a visual representation of the magmatic addition rates over time in the Mesozoic SCB (Fig. 2b).

## Data availability

The geochemical, isotope and geochronology data generated in this study are provided in the supplementary information files. Geochemical data for the Sierra Nevada Batholith shown in Fig. 3 were compiled from the NAVDAT database (https://www.navdat.org/).

## Code availability

This manuscript uses Iolite 4.8.1 to process raw LA-ICPMS data[68]. A free trial version of the software can be downloaded here: https://iolite.xyz/. Data collected at the USGS-Stanford Ion Microprobe Laboratory were reduced with the Excel 2003-based add-in Squid 2.51[56]. Calamari is a reimplementation of the code and can be accessed here: https://github.com/CIRDLES/Calamari.

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

## Acknowledgements

We thank Matt Coble, John Wiesenfeld and Zhan Peng for assistance with zircon and titanite petrochronology and Ilya Bindeman for stable isotope thermometry. Gabriel Romero and Ben Conway provided assistance in the field and in the laboratory. John Banacky and Ron Goodman of the Cucamonga Foothill Preservation Alliance, Alina Tibaldi (Rio Quarto University, Argentina) and Justin Okin provided assistance with field logistics. Financial support for this project was provided by the National Science Foundation grant EAR–2138733 (J.J.S. and E.A.M.), EAR–1655152 (Cecil and J.J.S.), EAR–0948706 and OCE–1338842 (J.S.L.), NSF–EAR 2138734 (K.A.K.), and Southern California Earthquake Center grants #21140 and #19023 (E.A.M. and J.J.S.). F.R. thanks the Geological Society of America for financial assistance.

## Author contributions

J.J.S., J.S.L., E.A.M., and K.A.K. conceived the study. Field work and sample collection was conducted by J.J.S., J.S.L., E.A.M., K.A.K., G.M-K., F.R., and J.D.B. Analyses and interpretations of the data were conducted by J.J.S., J.S.L., F.R., and J.D.B. Financial support was acquired by J.J.S., J.S.L., E.A.M., and K.A.K. The initial draft of the manuscript was written by J.J.S. and all authors contributed to editing and revisions.

## Competing interests

The authors declare no competing interests.
