## [Peer Review File · Nature Communications]

Magmatic Surge Requires Two-Stage Model for the Laramide OrogenyREVIEWER COMMENTS

Reviewer #1 (Remarks to the Author):

The research presented here has important implications for the driving mechanism for the Laramide orogeny. This major tectonic event in the western North America Cordillera has long been thought to have resulted from the subduction of an oceanic plateau and associated flat slab subduction. This flat slab is postulated to have resulted in the cessation of magmatism in the Sierra Nevada batholith and subsequent inboard migration of the Cordilleran arc. The authors convincingly show that magmatism occurred at a vigorous pace from 90-75 Ma for at least the Southern California batholith portion of the North American Cordilleran arc. They then propose an interesting two-state orogeny, and only the latter stage is inferred to have been related to flat slab subduction.

A few recent papers have noted problems with flat slab subduction at 90 Ma, as mentioned in the manuscript. These papers include ones on the timing of basin development in Montana and the recent (Tikoff et al., 2023) update of the “hit and run model” of outboard terranes colliding and then being translated northward along the North American margin. Schwartz et al. in the current manuscript provide by far the best documented and data-rich refutation of the classic flat slab model at 90 Ma, and focus on the area from which the model was developed.

I am not a geochronologist or isotope geochemist, but the data seem reproducible. Much of the methodology is standard practice and many of the authors have a long history of publishing high-quality geochronological, zircon trace element, and oxygen isotopic data.

I have only two suggestions for improvement. The authors could be clearer on explaining why there is an abrupt transition at ca. 85 Ma between the Sierra Nevada, where magmatism ceased, and the Southern California batholith, where the authors document a magmatic flare-up. It is interesting that the Southern California batholith has a decrease in magmatic tempo between ca. 118 Ma and 90 Ma, an interval marked by voluminous magmatism in the Sierra Nevada batholith. The same issue applies to the south with the Peninsular Ranges. I also did not get a clear understanding of what drove the surge in the Southern California batholith.

In summary, the authors clearly document that magmatism and a hot deep crust marked part of the interval when flat slab subduction was proposed to have occurred in the region of the Southern California batholith. The manuscript has significant implications for Cordilleran orogenesis and will also be of interest to many workers evaluating tectonic processes responsible for Pampean- and Laramide-type deformation. The paper is also likely to stimulate debate amongst the community of Cordilleran tectonicists.

Reviewer #2 (Remarks to the Author):

See attached.

Reviewer #3 (Remarks to the Author):

Review of Schwartz et al.

This is going to be a great paper. I don't think it is quite there, yet.

The authors make this basic observation: There is far too much magmatism and heating in the lower crust in the mid-Cretaceous for there to be a shallow slab below this segment of the arc. The authors have compelling data for this claim. This is a major problem for Cordilleran tectonics, and hence I think this paper is a significant contribution. They call on what I will call "local subduction": A subduction zone occurring only in southern California.

The writing is very clear and the figures are very good (this might have been the best manuscript that I've ever reviewed for the clarity of the writing – kudos to the authors).

Now, here are the troubles. The authors claim that if this area is hot with lots of deformation, it must be a magmatic arc and hence a subduction zone. This is not the only option and moreover there are problems with a subduction interpretation. First, their invoking of the Shatsky conjugate has major problems. The authors have made a mistake and this must be fixed before it can be published. Second, the authors avoid dealing with the issues rather than addressing them head-on related to a "local subduction model". They propose a model without adequately discussing why subduction cannot be happening to the south or the north. Fourth, there is just a bit too much twisting if the data to match the model – the authors need to be more consistent with the reported data on the Laramide and Sevier timing. Further, they give lip service to multiple interpretations in the introduction, but then don't seriously address. This is a problem: Their observations are far more consistent with a collisional model. These problems are all easily addressable and, if done, this would make a major contribution.

In a bulleted and elaborated list.

1. The conjugate Shatsky plateau argument does not make sense for two separate reasons. Torsvik et al. (2019) provide a recent review of the Shatsky and Hess rises. They note that the Shatsky rise experiences a series of eastward “ridge jumps” – movement of the spreading center between Pacific (west) and Farallon (east) – resulted in almost all of the plume-related volcanism associated with the Shatsky Rise being transferred to the Pacific plate. For this reason, the so-called Shatsky conjugate likely never existed (see Tikoff et al., 2023). I would point out that your data is a beautiful demonstration that this is likely true.

The second reason it is that you cannot delay when a plateau will hit a continent. That is, if there is nothing between middle of the Pacific ocean and North America (which is almost certainly incorrect – see below), it must hit at ~95 Ma. To call on it hitting at ~70 Ma is not permissible because of the plate motions require that it would be some distance across the coterminous US at that point. The authors just made a mistake. You do, however, have an easy substitute: The Hess plateau. The conjugate Hess plateau has the advantage that it might actually exist and it would hit about the right time and in the right place (with nothing in the way). So, if you are going to call on shallow subduction starting at 70 Ma, consider the conjugate Hess plateau.

But, I will point out the obvious: You have no reason to call on a shallow slab. The Sevier is over by then and so is much of the Laramide. We all have “slab on the brain” because we were taught that Laramide = shallow slab. That is not necessarily so – it is just a model. No shallow slabs are needed for an orogeny.

2. I think the authors are being a bit disingenuous about their model, by not addressing a major problem with it that they know well. The Southern California batholith sits between the Sierra Nevada batholith and the Peninsular Ranges batholith. Both of these larger arcs are off at 85 Ma using the conventional U-Pb zircon dating on granites (or 100 Ma, if you accept the Hildebrand and Whalen arc vs “slab breakoff” granites, the latter of which is a terrible term for what H & W are trying to describe). The point is that the authors are calling on a long-lived subduction zone to occur ONLY in the segment between the two larger batholiths – but are completely opaque about it. This must be said clearly.

In fact, the reason that Dickinson and Snyder (1979) proposed a shallow slab is because the coastal magmatic arcs have turned off. And, you can't have a shallow slab underneath the Sierra Nevada after 85 Ma because of the “arclogite” rocks preserved there. The presence of those rocks as xenoliths is why Saleeby (2003) put the shallow slab underneath the Mojave (your Southern California batholith). However, you have beautifully demonstrated that his model is also not viable.

So, what you are really calling on is a small segment of the Cordillera (southern California) having a subduction zone, but there is no subduction zone anywhere else. I can think of no modern analogy for that scenario. And, that subducting segment would be stationary through time. That is a MAJOR problem because this was demonstrably a time of oblique movement. If there is any obliquity (likely right-lateral), it would require that the northern tip of the subducting plate would move north, similar to the northward migration of the Mendocino triple junctions. That would move the subduction system underneath the Sierra Nevada were there is demonstrably no subduction. In short, the magmatic arc model doesn't work for 100 (90) -70 Ma.

And, if you are going to make this model, what type of plate interaction is going on to the north? To the south? When you start thinking in this way, it does not really work.

3. Another reason why your magmatic arc model doesn't work: It is not correlated with other magmatic provinces. Every other arc (both north and south of your area) in the US West shows a flareup at 100-90 Ma. Yours is later. It couldn't have been a normal arc. And, you never really talk about the two-mica granite belt, which I think you are including in your Southern California batholith. The two-mica granite belt happened from Canada to Mexico, including areas that demonstrably had no subduction. Therefore, subduction cannot be the cause for it under southern California, but not elsewhere. Again, there needs to be some acknowledgment of this problem.

4. You are cherry-picking your data for the age of the Sevier & Laramide orogeny is older than you state. Sevier and Laramide start at 100 Ma in SW Montana. The Sevier starts locally at 125 Ma, but really turns on at 100 Ma. A recent review paper by A. Yonkee and E. Balgord has a review of the timing. The Laramide also starts ~80 Ma, and locally earlier. You cite only the youngest ages, whereas most of the deformation is earlier than you cite. A recent review of the Laramide by Weil and Yonkee (2023) really should be cited.

In short, the deformation in the western US is much earlier than you call on shallow slab subduction. If you are connecting you post 70 Ma shallow slab subduction to Sevier-Laramide deformation, your model fails.

5. There are clear, consistent, and robust paleomagnetic data that the Insular superterrane +- other stuff is offshore all of California and Mexico at this time (~90 Ma). If you don't buy the paleomagnetic data, the detrital zircon data of Sauer et al. (2019) say the same thing. The Swakane gneiss of the North Cascades in Washington has nearly the identical detrital zircon signature as the Pelona-Orocopia schist in California at 75 Ma. That means the North Cascades (and the rest of the Insular terrane) was also there. Clennett et al. (2021) – based on seismic tomography - also has a GPlates model for this time. It says the same thing (although with westward subduction). The most likely option is that the Insular superterrane (+-) is adjacent to the California, but it could be offshore. Since the Insular terrane has an

arc, there isn't a slab left to subduct beneath southern California. Moreover, the Insular terrane is blocking the speculated Shatsky rise (which probably doesn't exist) from hitting southern California.

These are all reasons why your subduction model as presented is probably incorrect. I don't think it is possible.

Yet, I really like this paper.

I am going to be completely self-serving here, but at least I'll admit it up front. A subduction model cannot work for your observations. On the other hand, a collisional model can produce exactly the features you see at exactly the time that you see them. You cite the hit-and-run model (Maxson and Tikoff, 1996) and the idea of a collision as an option, but you never seriously address it. There is no reason why you shouldn't address it – it is an idea in the literature (also see Tikoff et al., 2023). You might also want to cite Hildebrand and Whalen's (2022) paper on the Peninsular Ranges orogeny: I neither like the name nor accept the idea of westward subduction, but the point is that it also calls for a collision at 100 Ma exactly where your data is from.

Why is a collision model compelling? First, it exactly explains your observations. Collisions produce granites, particularly in areas that were recently magmatic arcs. The collision would take place at 100 Ma. It takes 10-15 m.y. to produce crustal melting associated with crustal thickening, which is exactly when you see your magmatic flare up. You are also seeing hot lower crustal rocks, which is exactly what is formed in collisional zones. Second, it is predicted to occur exactly where your deformation occurs. And, it would not move because it is tied to the geometry of the continent, not to a migrating triple junction (e.g., Mendocino-like). Third, it gets around all the problems associated with trying to sneak a small subduction zone into southern California.

Even if your "local subduction" model worked, it would be incumbent on you to at least mention that a collision is possible. Then, you could say why you like one model over the other. To me, that is good science. Given, however, that the collision model works much better than a subduction model, it really should be discussed (or, heavens forbid, adopted).

Basil Tikoff

Reviewer #1 (Remarks to the Author):

The research presented here has important implications for the driving mechanism for the Laramide orogeny. This major tectonic event in the western North America Cordillera has long been thought to have resulted from the subduction of an oceanic plateau and associated flat slab subduction. This flat slab is postulated to have resulted in the cessation of magmatism in the Sierra Nevada batholith and subsequent inboard migration of the Cordilleran arc. The authors convincingly show that magmatism occurred at a vigorous pace from 90-75 Ma for at least the Southern California batholith portion of the North American Cordilleran arc. They then propose an interesting two-state orogeny, and only the latter stage is inferred to have been related to flat slab subduction.

A few recent papers have noted problems with flat slab subduction at 90 Ma, as mentioned in the manuscript. These papers include ones on the timing of basin development in Montana and the recent (Tikoff et al., 2023) update of the “hit and run model” of outboard terranes colliding and then being translated northward along the North American margin. Schwartz et al. in the current manuscript provide by far the best documented and data-rich refutation of the classic flat slab model at 90 Ma, and focus on the area from which the model was developed.

I am not a geochronologist or isotope geochemist, but the data seem reproducible. Much of the methodology is standard practice and many of the authors have a long history of publishing high-quality geochronological, zircon trace element, and oxygen isotopic data.

I have only two suggestions for improvement. The authors could be clearer on explaining why there is an abrupt transition at ca. 85 Ma between the Sierra Nevada, where magmatism ceased, and the Southern California batholith, where the authors document a magmatic flare-up.

We have revised the manuscript to more clearly discuss the distinct geologies in the SNB and the SCB. We follow Saleeby (2003) in arguing that the California arc is segmented at the latitude of the SNB-SCB junction, or at the future Garlock fault. In our view, arc magmatism continued south of the SNB, in the SCB, eastern PRB and northern Mexico and this can be explained by segmentation for the arc. We also discuss the hit and run model (Tikoff et al., 2023) and incorporate it into our revised model (see Figure 5).

It is interesting that the Southern California batholith has a decrease in magmatic tempo between ca. 118 Ma and 90 Ma, an interval marked by voluminous magmatism in the Sierra Nevada batholith. The same issue applies to the south with the Peninsular Ranges.

Yes, we noticed this too in our data. It's unclear if this is related to preservation of the arc in the SCB or a real magmatic lull. For example, Jacobson et al. (2011) suggested that this older part of the arc might have been displaced by sinistral strike slip movement. Our data do not support either model, so we do not comment on it at present, but it is a great observation.

I also did not get a clear understanding of what drove the surge in the Southern California batholith.

This is a great point. The data we have presently do not allow us to fingerprint the cause of the magmatic surge. There are a variety of models that have been put forward in the Southern Sierra Nevada batholith and the Peninsular Ranges batholith; however, there has yet to be a comprehensive isotopic study of the SCB. We are currently working on this problem and hope to have results in the next few years. Stay tuned.

In summary, the authors clearly document that magmatism and a hot deep crust marked part of the interval when flat slab subduction was proposed to have occurred in the region of the Southern California batholith. The manuscript has significant implications for Cordilleran orogenesis and will also be of interest to many workers evaluating tectonic processes responsible for Pampean- and Laramide-type deformation. The paper is also likely to stimulate debate amongst the community of Cordilleran tectonicists.

Thank you! We hope that it will lead to debate and better understanding of the Laramide orogeny as well.

Review of NCOMMS-22-52480-T, “Magmatic surge requires two-stage model for the Laramide orogeny”

Summary:

This study presents a compilation of new igneous and metamorphic zircon and titanite UPb dates, areal addition rates, quartz-garnet oxygen isotope thermometry, and regional mapping from the Southern California Batholith to reconstruct the magmatic and deformation history of the region. In turn, the dataset directly tests the timing of the proposed collision of the conjugate Shatsky rise with North America and development of a flat slab, which is the leading hypothesis cited for inboard thick-skinned, Laramide orogenesis across the western U.S. The paper presents four clear and highly citable figures, especially figure 2. Due to the nature of the data, much of it is in the supplementary material, but if journal restraints allow, I recommend adding some field or petrographic photos that highlight the structural deformation of the region. The major take-away and contribution of this paper is stated at Lines 196-197: “These relationships coupled with our data from the SCB lead us to the conclusion that the Laramide orogeny cannot have a single driving mechanism”. The data support this conclusion, and the integration of existing datasets supports that the initial phase of thick-skinned involved deformation across the western U.S. cannot be driven by subduction of a conjugate Shatsky Rise as the timing and projected path is at odds with data from southern California all the way to Montana. For these reasons, I recommend the paper be considered for publication with Nature Communications after moderate revision. My comments below are intended to help the authors to make it clear from the start that this is a regional study in southern California, but, as they demonstrate, it links to the broader Laramide system.

Specific Comments: Please see the attached PDF for additional queries.

Line 23: which arc? southern Sierra Nevada? Southern California batholith to the south? I recommend adding this detail as there is no geographic information in the abstract to indicate the location of this study.

We have clarified that we mean the Southern California Batholith. We have also added more geographical information to the abstract.

Lines 27-28: At the latitude of southern California. Given the complexity of this mountain building across the western US, adding some geographic context will help place this study within the broader picture.

We have expanded this sentence to clarify that we are referring to southern California sector of the western North American Cordillera

Line 36: likely 60 myr in Montana and Wyoming. Perhaps consider a range?

We have made this a range as suggested.

Line 69: I recommend citing the number from existing literature and new, since the new data is a significant chunk of the compilation!

Done.

Line 76: Supplementary Table 8

Done.

Lines 77: This "pulse" is not well-defined and could easily be interpreted as magmatism during a "lull", which of course doesn't always have to have complete shut-off of the arc. I recommend separating these numbers from the three main pulses as a lull or minor pulse.

Good point. We have removed this pulse from the sentence.

Line 99: "cool" finding.

Thanks!

Lines 101-108: I recommend adding a few photos to the supplementary, especially a map that indicates where some of these shear zones are located. At present, this data is not included in the manuscript and Figure 1 lacks detail to identify where these are located for at least everything south of Nacimiento.

We have added two new figures to the Supplementary file as suggested. The first is a map showing the present-day locations of the shear zones, and the second is a palinspastic reconstruction showing their location in the Late Cretaceous. We also added a figure showing field photos of the ductile shear zones.

Line 145: is interpreted as

Done.

Line 156: Timing in Wyoming okay--Winds are ~ 70 Ma, but the sed record and low-T thermochron in Montana places basement involved deformation earlier >80 Ma. I recommend revising this to state that there was an increase in exhumation by 70 in the northernmost Laramide (e.g., see Ronemus et al. 2023, Beartooths--shameless plug, but the data support earlier onset in Montana-Wyoming). Grouping Montana with Utah and Colorado is troublesome given data post this 2015 paper.

We have revised this section and cited Ronemus et al. (2023).

Line 191: Yes! This addition is key to testing the Shatsky model, which doesn't work in southwest Montana. See above comment at Line 156. I recommend simply removing northern Wyoming and southern Montana (Wyoming included given Ronemus et al. 2023 study).

Done.

Line 203: at this latitude

We have revised the sentence.

Lines 203-204: How does this lead to deformation in Wyoming and Montana? I few words here would strengthen this argument.

We appeal to the 'hit-and-run' model to explain deformation in Wyoming and Montana. We have developed this more in the revised section (A two-stage model for the Laramide orogeny).

Abstract: I recommend adding geographic information to the abstract as the study is specific to magmatic and metamorphic rocks preserved in southern California. Although the study is located where the Shatsky Rise is predicted to collide, it is crucial to place this work in the broader context of the Laramide, which is much different in style and timing farther north.

Good points. We have added additional geographical information to the abstract to clarify locations.

Deformation: The paragraph from Lines 100-118 is a crucial field check and nice addition to the geochronology, but it left me wondering about where all these shear zones are in relation to the rocks dated. As stated at Lines 100-108: I recommend adding a few photos to the supplementary, especially a map that indicates where some of these shear zones are located. At present, this data is not included in the manuscript and Figure 1 lacks detail to identify where these are located for at least everything south of Nacimiento.

Great suggestion. We have added a new map (Supplementary Figure 1) that shows the locations of the shear zones in present-day and in a Late Cretaceous palinspastic reconstruction. We've also added a new figure showing field photos of the shear zones (Supplementary Figure 2).

Broad extraction of results to the western US: Building upon what the authors have written, I have left a few comments in places where it needs to be clear what is happening in southern California at the same time as elsewhere (e.g., Utah vs Montana). Specifically, how does regional transpression along the margin lead to deformation in the Northern Rockies? A big question, but one worth commenting on, even if building off Tikoff's arguments (See line 204).

Good point. Our data bear on this point and indicate that the conjugate Shatsky was not a driver. Therefore another mechanism is required. We now discuss the Tikoff et al. (2023) 'hit-and-run model' which provides a mechanism for deformation in the northern Rockies.

I encourage the authors to contact me if any of the above points are not clear.

Thank you, Devon.

Best Wishes,

Devon A. Orme

Reviewer #3 (Remarks to the Author):

Review of Schwartz et al.

This is going to be a great paper. I don't think it is quite there, yet.

The authors make this basic observation: There is far too much magmatism and heating in the lower crust in the mid-Cretaceous for there to be a shallow slab below this segment of the arc. The authors have compelling data for this claim. This is a major problem for Cordilleran tectonics, and hence I think this paper is a significant contribution. They call on what I will call "local subduction": A subduction zone occurring only in southern California.

The writing is very clear and the figures are very good (this might have been the best manuscript that I've ever reviewed for the clarity of the writing – kudos to the authors).

Now, here are the troubles. The authors claim that if this area is hot with lots of deformation, it must be a magmatic arc and hence a subduction zone. This is not the only option and moreover there are problems with a subduction interpretation. First, their invoking of the Shatsky conjugate has major problems. The authors have made a mistake and this must be fixed before it can be published. Second, the authors avoid dealing with the issues rather than addressing them head-on related to a "local subduction model". They propose a model without adequately discussing why subduction cannot be happening to the south or the north. Fourth, there is just a bit too much twisting if the data to match the model – the authors need to be more consistent with the reported data on the Laramide and Sevier timing. Further, they give lip service to multiple interpretations in the introduction, but then don't seriously address. This is a problem: Their observations are far more consistent with a collisional model. These problems are all easily addressable and, if done, this would make a major contribution.

Thank you, Basil. We have addressed all of your points in this revision and discuss them point by point below. We appreciate the suggestions and comments as they have truly improved the science of the manuscript.

In a bulleted and elaborated list.

1. The conjugate Shatsky plateau argument does not make sense for two separate reasons. Torsvik et al. (2019) provide a recent review of the Shatsky and Hess rises. They note that the Shatsky rise experiences a series of eastward "ridge jumps" – movement of the spreading center between Pacific (west) and Farallon (east) – resulted in almost all of the plume-related volcanism associated with the Shatsky Rise being transferred to the Pacific plate. For this reason, the so-called Shatsky conjugate likely never existed (see Tikoff et al., 2023). I would point out that your data is a beautiful demonstration that this is likely true.

This is a great point. We no longer refer to the conjugate Shatsky Rise and cite Torsvik et al. (2019). We agree completely with this point and now mention the possibility that the conjugate Shatsky Rise likely never existed. Thanks for making this point.

The second reason it is that you cannot delay when a plateau will hit a continent. That is, if there is nothing between middle of the Pacific ocean and North America (which is almost

certainly incorrect – see below), it must hit at ~95 Ma. To call on it hitting at ~70 Ma is not permissible because of the plate motions require that it would be some distance across the coterminous US at that point. The authors just made a mistake. You do, however, have an easy substitute: The Hess plateau. The conjugate Hess plateau has the advantage that it might actually exist and it would hit about the right time and in the right place (with nothing in the way). So, if you are going to call on shallow subduction starting at 70 Ma, consider the conjugate Hess plateau.

You are correct about the delay. This is an error, but you make a great point about the Hess. We still favor a shallow slab model for southern California arc segment and we have taken your suggestion to substitute the conjugate Shatsky for the conjugate Hess plateau. Another great point. Thanks.

But, I will point out the obvious: You have no reason to call on a shallow slab. The Sevier is over by then and so is much of the Laramide. We all have “slab on the brain” because we were taught that Laramide = shallow slab. That is not necessarily so – it is just a model. No shallow slabs are needed for an orogeny.

We agree that a shallow slab model is not required to explain geology north of the SCB, but an oceanic plateau collision after 75-70 Ma in the SCB segment does explain a great deal of the geology south of the SNB. For one, thick-skinned deformation really kicks off after 75 Ma in the Laramide foreland and collision with the Hess conjugate can explain the timing and geometry of thrusts. Second, the POR schists were not deposited until after 75 Ma, and then they have to be metamorphosed and underthrust beneath the SCB. Xenoliths beneath the Colorado plateau also record 100s of kms of lateral displacement of the sub-arc lithosphere after 75 Ma. The Hess conjugate collision model after 75-70 Ma still does a good job explaining these features. In our revision, we also emphasize that the Mesozoic California arc is segmented at the present-day Garlock fault, and we propose that Hess conjugate collision occurred simultaneously with the ‘run’ phase, which we now highlight this in the manuscript (see revised Fig. 5).

2. I think the authors are being a bit disingenuous about their model, by not addressing a major problem with it that they know well. The Southern California batholith sits between the Sierra Nevada batholith and the Peninsular Ranges batholith. Both of these larger arcs are off at 85 Ma using the conventional U-Pb zircon dating on granites (or 100 Ma, if you accept the Hildebrand and Whalen arc vs “slab breakoff” granites, the latter of which is a terrible term for what H & W are trying to describe). The point is that the authors are calling on a long-lived subduction zone to occur ONLY in the segment between the two larger batholiths – but are completely opaque about it. This must be said clearly.

In fact, the reason that Dickinson and Snyder (1979) proposed a shallow slab is because the coastal magmatic arcs have turned off. And, you can't have a shallow slab underneath the Sierra Nevada after 85 Ma because of the “arclogite” rocks preserved there. The presence of those rocks as xenoliths is why Saleeby (2003) put the shallow slab underneath the Mojave (your Southern California batholith). However, you have beautifully demonstrated that his model is also not viable.

So, what you are really calling on is a small segment of the Cordillera (southern California)

having a subduction zone, but there is no subduction zone anywhere else. I can think of no modern analogy for that scenario. And, that subducting segment would be stationary through time. That is a MAJOR problem because this was demonstrably a time of oblique movement. If there is any obliquity (likely right-lateral), it would require that the northern tip of the subducting plate would move north, similar to the northward migration of the Mendocino triple junctions. That would move the subduction system underneath the Sierra Nevada were there is demonstrably no subduction. In short, the magmatic arc model doesn't work for 100 (90) -70 Ma.

And, if you are going to make this model, what type of plate interaction is going on to the north? To the south? When you start thinking in this way, it does not really work.

We agree and disagree with these points. Basil's comments apply beautifully to rocks north of the Garlock fault; however, south of the Garlock, magmatism didn't shut off and there is abundant evidence to support continued magmatism which overlaps with that in the SCB. The point about the chemistry of the arc rocks is also contentious. We treat each of these points separately below:

1) We agree that magmatism was waning and shutting down in the SNB around 85. However, magmatism did not shut off at 85 Ma south of the Garlock fault. There is abundant evidence for coeval magmatism south of the SCB in southern California and northwest Mexico. For example, 1) there is evidence for continued eastward migration of Late Cretaceous plutonism into the upper-plate zone of the adjacent PRB (85-80 Ma: Premo et al. 2014), 2) there is widespread plutonism in the Sonora batholith from 91 to 50 Ma with peak activity at 71 Ma, and 3) there is a long-lived record of magmatism in the Sinaloa batholith from 100 to 45 Ma. So, the point about magmatism shutting down in SNB is correct, but it is not correct for arc segments south of the Garlock fault. Also see Fitz-Diaz et al. (2018) for a review of NW Mexican geology and magmatism. Their Figure 14 illustrates continuous magmatism in this southern segment of the margin from 90-50 Ma.

2) The point about the SCB rocks being 'collisional' and related to slab break off is based on trace-element discrimination schemes from Hildebrand and Whalen (2001). These are problematic for a variety of reasons. First, the primary problem with the Hildebrand and Whalen diagrams is that they rely on trace element compositions which are primarily a function of source and crystallization histories. Tectonic environment is secondary but this is what they are trying to emphasize. The crux of their problem is that their discrimination schemes rely on 'Sr/Y' as a tectonic indicator for slab-break off. Our working group has shown that Sr/Y is deceptive and can be related to crystal accumulation, particularly in high volume magma systems like the La Posta. Brackman and Schwartz (2022) showed this for clinopyroxene geochemistry and Carty et al. (2021) showed this for amphiboles in a high-volume magmatic system in Zealandia, and Brackman (MS thesis, 2022) showed this for the La Posta (unpublished right now). In all three studies, we can demonstrate that amphibole and clinopyroxene Sr/Y values do not match bulk-rock values, and this is best explained as the result of crystal accumulation of plagioclase in granitoids. This suggestion is supported by a number of recent studies that argue that plutons are cumulates and do not

represent liquid compositions (see Schaen et al. 2018; Barnes et al. 2019; Werts et al. 2020). Therefore, we do not place much faith in bulk-rock trace element discrimination ratios and urge caution in applying them to interpret the tectonic history of Cordilleran granites.

3. Another reason why your magmatic arc model doesn't work: It is not correlated with other magmatic provinces. Every other arc (both north and south of your area) in the US West shows a flareup at 100-90 Ma. Yours is later. It couldn't have been a normal arc. And, you never really talk about the two-mica granite belt, which I think you are including in your Southern California batholith. The two-mica granite belt happened from Canada to Mexico, including areas that demonstrably had no subduction. Therefore, subduction cannot be the cause for it under southern California, but not elsewhere. Again, there needs to be some acknowledgment of this problem.

Indeed, this is abnormal arc behavior in the classical sense, although it is well known in the arc community that the timing of flare-ups varies along strike within continental arcs and they can occur within a ~30 Myr window, and can even sometimes shift back into earlier plutonic domains. This comment motivated us to highlight more of the magmatic history to the south of the SCB, in Sonora and other areas to the south, as evidence of enduring arc activity. In this light, there is nothing unusual about the SCB flare-up being 5-15 Myr younger than in the Sierra Nevada and Peninsular Ranges, which are in fact the outboard segments of the arc. In fact, peak magmatism in the Sonora batholith occurred at 71 Ma, which is 10 Myr younger than in the SCB. The point is that magmatic flare-ups do not occur at the same time in adjacent segments of arcs. There is a nice review of this by Paterson and Ducea (2015) and Kirsch et al. (2016) present a larger data compilation from the entire North and South America Cordillera. We now discuss these points in the zircon geochronology section of the results section.

We agree with Basil about the petrogenesis of the two-mica granite belt/anatectic belt, but these rocks are different than the SCB. The two-mica granite belt/anatectic belt rocks are dominantly 70 wt.% SiO₂ peraluminous leucogranites and are chemically distinct from the SCB rocks which range from gabbro to granite. Hence, they are not included in the SCB and we make that point in the revised version. To illustrate this point, we've added a new figure (Fig. 3) which shows the geochemistry of the SCB against arc rocks of the SNB (120-85 Ma), global melt inclusion data, and the Cordilleran anatectic belt/2-mica granite belt. The plots show that the SCB rocks are exactly like those in the Early to Late Cretaceous SNB, and both groups of arc rocks are distinct from likely collisional rocks in the Cordilleran anatectic belt.

4. You are cherry-picking your data for the age of the Sevier & Laramide orogeny is older than you state. Sevier and Laramide start at 100 Ma in SW Montana. The Sevier starts locally at 125 Ma, but really turns on at 100 Ma. A recent review paper by A. Yonkee and E. Balgord has a review of the timing. The Laramide also starts ~80 Ma, and locally earlier. You cite only the youngest ages, whereas most of the deformation is earlier than you cite. A recent review of the Laramide by Weil and Yonkee (2023) really should be cited.

Fair points and we have revised the timing we cite. We also cite Weil and Yonkee (2023).

In short, the deformation in the western US is much earlier than you call on shallow slab subduction. If you are connecting you post 70 Ma shallow slab subduction to Sevier-Laramide deformation, your model fails.

Basil makes a good point about this, and his “hit and run” model likely accounts for deformation in the northern US Cordillera prior to 75 Ma. We now discuss the ‘hit’ phase as likely accounting for early deformation and we emphasize that early deformation could not have been related to a collision with the conjugate Shatsky Rise (one of the main points of our manuscript). We also show these features in our revised model in Figure 5.

5. There are clear, consistent, and robust paleomagnetic data that the Insular superterrane +- other stuff is offshore all of California and Mexico at this time (~90 Ma). If you don't buy the paleomagnetic data, the detrital zircon data of Sauer et al. (2019) say the same thing. The Swakane gneiss of the North Cascades in Washington has nearly the identical detrital zircon signature as the Pelona-Orocopia schist in California at 75 Ma. That means the North Cascades (and the rest of the Insular terrane) was also there. Clennett et al. (2021) – based on seismic tomography - also has a GPlates model for this time. It says the same thing (although with westward subduction). The most likely option is that the Insular superterrane (+-) is adjacent to the California, but it could be offshore. Since the Insular terrane has an arc, there isn't a slab left to subduct beneath southern California. Moreover, the Insular terrane is blocking the speculated Shatsky rise (which probably doesn't exist) from hitting southern California.

The Baja-BC model works for rocks north of the Garlock fault; however, the kinematics of Late Cretaceous shear zones from 83-70 in the SCB show sinistral-reverse sense of shear, and this motion would send the Insular terranes southward (the opposite direction for the Baja BC model). To make this point clearer, we now show field photos of these sinistral shear zones in a new figure in the Supplementary File). For this reason, we have difficulty calling upon an Insular terrane collision in the SCB. It may be that they were offshore and docked north of the Garlock fault, but we see no evidence for their collision with the SCB and the predicted kinematics do not match our observations.

Another issue with a ‘hit’ in the SCB is that the timescales of melting the lower crust of the SCB do not support a collisional model at 100 Ma (see Basil's point below). For example, thermal models suggest that maximum temperatures in the middle to lower crust are attained 40–60 Myr after crustal thickening (England and Thompson, 1984, 1986; Clark et al., 2011), ruling out Insular collision at 100 Ma as causing crustal anatexis in the SCB at 85-70 Ma. Timescales are simply too short. And there is good evidence that magmas were sourced from mantle melting (see Barth et al., 2016 for Big Bear rocks). The 100 Ma Insular collision does work for melting the crust in the hinterland and this is a good explanation for the origin of the Cordilleran Anatectic Belt/two-mica belt (which Basil mentions in Tikoff et al. 2023).

Lasty, Basil mentions the Pelona-Orocopia schists, which are also a problem for a collisional model south of the Garlock fault. For one, the Pelona schist was deposited after 75 Ma, and then experienced moderate to high-P low T metamorphism after 60 Ma (Jacobson et al. 2011). Chapman et al. (2020) also showed that arc xenoliths were also translated >500 km towards to the Colorado plateau after 70 Ma. These features are best described by a shallow subduction model at the latitude of the SCB. Given sinistral motions at this time, the ‘run’ model can’t work south of the Garlock fault. However, we acknowledge that dextral strike-slip motion after 70 Ma is permissible but there are no known faults in the SCB that record this motion.

These are all reasons why your subduction model as presented is probably incorrect. I don't think it is possible.

Yet, I really like this paper.

I am going to be completely self-serving here, but at least I'll admit it up front. A subduction model cannot work for your observations. On the other hand, a collisional model can produce exactly the features you see at exactly the time that you see them. You cite the hit-and-run model (Maxson and Tikoff, 1996) and the idea of a collision as an option, but you never seriously address it. There is no reason why you shouldn't address it – it is an idea in the literature (also see Tikoff et al., 2023). You might also want to cite Hildebrand and Whalen's (2022) paper on the Peninsular Ranges orogeny: I neither like the name nor accept the idea of westward subduction, but the point is that it also calls for a collision at 100 Ma exactly where your data is from.

Why is a collision model compelling? First, it exactly explains your observations. Collisions produce granites, particularly in areas that were recently magmatic arcs. The collision would take place at 100 Ma. It takes 10-15 m.y. to produce crustal melting associated with crustal thickening, which is exactly when you see your magmatic flare up. You are also seeing hot lower crustal rocks, which is exactly what is formed in collisional zones. Second, it is predicted to occur exactly where your deformation occurs. And, it would not move because it is tied to the geometry of the continent, not to a migrating triple junction (e.g., Mendocino-like). Third, it gets around all the problems associated with trying to sneak a small subduction zone into southern California.

Even if your “local subduction” model worked, it would be incumbent on you to at least mention that a collision is possible. Then, you could say why you like one model over the other. To me, that is good science. Given, however, that the collision model works much better than a subduction model, it really should be discussed (or, heavens forbid, adopted).

Basil Tikoff

We really like the ‘hit-and-run’ model as it explains a lot of the geology north of the Garlock fault. While there are geologic problems with extension of the model to the SCB and southward into Mexico (see above), the ‘hit-and-run’ model works for rocks north of the SCB. We now discuss and incorporate the hit-and-run model into our heavily revised section “A two-stage model for the Laramide orogeny” and we show the model in our revised Figure 5. We very much appreciate the thoughtful comments

and criticism, and look forward to future spirited conversations about the Laramide orogeny.

We hope that you will find these revisions suitable and look forward to the opportunity to publish this work in Nature Communications.

REVIEWER COMMENTS

Reviewer #1 (Remarks to the Author):

The authors satisfactorily addressed my concerns and made significant improvements in response to Reviewer #3. Discarding the Shatsky plateau and adding the subduction of the Hess Rise makes the overall interpretations more logical. The addition of the plate models to Fig. 5, including showing the Hess Rise is also useful. I only have one substantive comment and the intent of the other comments is to improve the clarity.

I think the authors make a couple of erroneous statements about the timing of displacement of the western Idaho shear zone (see Giorgis et al., 2008 GSAB; Braudy et al., 2017). It was only active during the “hit phase” in the Tikoff et al. model. The way that the text reads on lines 266-268 is that the shear zone was active from 75-40 Ma. Also, on lines 136-138 about the timing of sinistral transpression in the region of the Southern California batholith, and dextral transpression in the western Idaho shear zone, it should be clearer that only the older southern California shear zones overlapped temporally with the Idaho shear zone.

78. I couldn't find the Alamo-Frasier Mountain block on Fig. 1B.

120. I suggest modifying the statement about “Our regional mapping of ductile shear zones.” Several excellent structural geologists are co-authors, but the evidence presented appears to be in the cited papers other than some field shots of shear zones in Supplementary Figure 2.

155. Add “central and” before “southern SNB” as the flare-up extends over a broader extent?

Fig. 5. The brown blob next to the word Pacific on the plate model for 85-75 Ma should be labeled.

Reviewer #2 (Remarks to the Author):

I have read the revised version this manuscript and find that it addresses all of my original concerns and suggestions. The additional text regarding alternative models to explain the timing and style of deformation strengthens the paper, as does the added lines to provide geographic context. I recommend the paper be accepted and its present form.

Reviewer #3 (Remarks to the Author):

Review of Schwartz et al.

I reviewed the first version of this paper.

The authors have done a very good job of revising the manuscript. At its core, the authors document that there is far too much magmatism and heating in the lower crust in the mid-Cretaceous for there to be a shallow slab below this segment of the arc until after 75 Ma. This dataset is pretty compelling for knocking out the prevailing model of shallow slab subduction in the middle Cretaceous. The writing is very clear and the figures are very good.

I feel that this manuscript is, in some ways, very different from the one that I originally reviewed. The authors have cleared up most of the major problems in the original version (e.g., Shatsky conjugate, dealing with the Insular superterrane southerly position, dealing with collisional models, etc.). In their revision, however, they have introduced a few new problems that are not necessary for the main point of this paper. The biggest problem is the last one in this list.

1. The authors did deal with the “local subduction” problem, but basically incorporating all of northern Mexico in the subduction system. I think you are stretching your claims here. The Peninsular Range magmatic arc is widely acknowledged to turn off at ~90 Ma. There is younger magmatism, but it is not clear how it relates to subduction. In short, by dealing with the Mexican subduction and magmatic history, you are making some interpretations that may not be correct. I read that literature, but I don't work there and it is hard for me to evaluate the competing models. The point for this manuscript is that it was on firmer ground when they were talking about the Southern California batholith. Please consider paring it down to what you have data about. Alternatively stated, the focus of the original manuscript was better on this issue.

2. I am surprised – and gratified – that you took on the hit-and-run model (which was surprisingly more strongly raised by another reviewer). But, because that was not in the original manuscript, it needs some revision.

a. The western Idaho shear zone is off at 85 Ma (at absolute latest). I pointed out this error in my scanned version of the text. In fact, none of the dextral transpressional shear zones in the arcs are active after ~85 Ma. The fault/shear zones that moves the Insular superterrane northwards are – I think – coastward from the western Idaho shear zone and the Sierra Nevada batholith.

b. I remember from Paul Umhoefer that most people put the Kula-Farallon triple junction at the southern tip of the Insular superterrane. To me, that is the best plate model, because the placement of the North America-Farallon-Kula triple junction is effectively unconstrained and it promotes the northward motion of the Insular superterrane. Your Figure 4 needs to reflect that position (you've adapted it from Yonkee and Weil, who don't really deal with the Insular superterrane). For the record, I think there is yet another plate – the Resurrection – in the mix between the Kula and Farallon plate, but you probably don't have to deal with it.

c. Your model for continued subduction in southern California is, in fact, at odds with the paleomagnetic data from Canada, NW Washington, and Alaska. The paleomagnetism has the southernmost part of the Insular superterrane half way down into Mexico at 100 Ma. That is fine that you are not fully compatible – you can have a different model. But, for scientific fairness, I think that you must acknowledge that your model is not consistent with it (and, for the record, my money is on the paleomagnetism being correct).

d. You unfairly criticize the hit-and-run model for not being able to deal with a subduction complex (represented by the Pelona-Orocopia schists). In fact, it is exactly compatible – as is the paleomagnetic data – with that interpretation. The Insular terrane must have moved past southern California by 75 Ma on its way northward (it is in place by 55 Ma). That scenario is clearly laid out in Tikoff et al. (2023) and is consistent with Sauer et al. (2019) correlation of the Swakane gneiss of Washington State with the southern California schists.

3. The single biggest problem with the manuscript is the inclusion of the sinistral-reverse shear zones present in the southern California batholith. These were absent from the original manuscript, and they should have stayed out of this version. There are no scientific papers that have documented those shear zones that have made it through peer review (that I know of or that are cited). I don't doubt that this team can determine kinematics of shear zones. But, that is not how science works. You can only use data if it is already published or is presented in the manuscript. Neither is applicable. I see exactly why the authors want to utilize it: It appears to make a better story. But, it is inappropriate for a scientific

manuscript to utilize unpublished data and I think that not presenting it is going to stop you from making a mistake.

a. There is a massive amount of vertical axis rotation in this area of California during the Cenozoic (and probably the Late Cretaceous). While you use the reconstruction of Powell (1991), you don't talk about the reconstruction of those shear zones into a Late Cretaceous position. Their orientation is critical. There is suspected sinistral deformation associated with dextral transpressional "hit" in Idaho, because it is not an orthogonal collision. Rather, the orientation of the boundary is going to control whether a shear zone is sinistral or dextral. If there is any place in the Cordillera where a sinistral-reverse zone will happen during the initial obliquely dextral collision of the Insular block, it is going to be in southern California.

b. There is no documentation of the timing of the shear zones, and the timing information is essential. You make a point that you think that the reverse-sinistral shear zones might record the initiation of the shallow slab subduction associated with Hess conjugate (I think that intuition is likely correct). But, it all hinges on their timing, which you DO NOT constrain. If they are ~75 Ma, they are almost certainly the result of trailing edge subduction.

c. Sinistral shear zones can accommodate dextral motion. One example is antithetic block rotation. A second example, as argued in Tikoff et al. (2023b), the sinistral motion on the Lewis and Clark line is a result of the larger dextral motion. That sinistral motion starts at ~85 Ma, which may be consistent with your timing.

The most important reason to not present information about the shear zones is that you don't need to. Your zircon data stands on its own as a completely compelling case for there not being shallow slab subduction until after 75 Ma (or never, in my opinion). You have great structural geologists on your team. Let them publish the sinistral-reverse story when they know: 1) What was the orientation of the shear zones when they were active; 2) When were they active; and 3) How it relates to the larger tectonic context. Besides, it gives your team a second major paper to write. You are – in my opinion – making an unforced error by trying to deal with the shear zones in this paper.

I completely accept your main point: There cannot be a shallow slab in southern California until after 75 Ma. Your data support that. I think your model is incorrect for continued subduction until 75 Ma in southern California, but I must admit that it is admissible with your data. Anything after 75 Ma is conjecture, and is not the strength of this manuscript.

Reviewer #1 (Remarks to the Author):

The authors satisfactorily addressed my concerns and made significant improvements in response to Reviewer #3. Discarding the Shatsky plateau and adding the subduction of the Hess Rise makes the overall interpretations more logical. The addition of the plate models to Fig. 5, including showing the Hess Rise is also useful. I only have one substantive comment and the intent of the other comments is to improve the clarity.

Thank you. We have made all of your suggested edits.

I think the authors make a couple of erroneous statements about the timing of displacement of the western Idaho shear zone (see Giorgis et al., 2008 GSAB; Braudy et al., 2017). It was only active during the “hit phase” in the Tikoff et al. model. The way that the text reads on lines 266-268 is that the shear zone was active from 75-40 Ma. Also, on lines 136-138 about the timing of sinistral transpression in the region of the Southern California batholith, and dextral transpression in the western Idaho shear zone, it should be clearer that only the older southern California shear zones overlapped temporally with the Idaho shear zone.

We agree that our statements were unclear. We have removed the paragraph that contains the sentence on lines 136-138 which was also suggested by Reviewer #3. We have also removed the sentence on lines 266-268.

78. I couldn't find the Alamo-Frasier Mountain block on Fig. 1B.

We've added a label for the Alamo-Frasier Mnt block in Fig. 1B.

120. I suggest modifying the statement about “Our regional mapping of ductile shear zones.” Several excellent structural geologists are co-authors, but the evidence presented appears to be in the cited papers other than some field shots of shear zones in Supplementary Figure 2.

We have removed this section as suggested by Reviewer 1 and 3. As they suggest, it is best to put this in another manuscript.

155. Add “central and” before “southern SNB” as the flare-up extends over a broader extent?

Done.

Fig. 5. The brown blob next to the word Pacific on the plate model for 85-75 Ma should be labeled.

Done.

Reviewer #2 (Remarks to the Author):

I have read the revised version this manuscript and find that it addresses all of my original concerns and suggestions. The additional text regarding alternative models to explain the timing and style of deformation strengthens the paper, as does the added lines to provide geographic context. I recommend the paper be accepted in its present form.

Thank you!

Reviewer #3 (Remarks to the Author):

Review of Schwartz et al.

I reviewed the first version of this paper.

The authors have done a very good job of revising the manuscript. At its core, the authors document that there is far too much magmatism and heating in the lower crust in the mid-Cretaceous for there to be a shallow slab below this segment of the arc until after 75 Ma. This dataset is pretty compelling for knocking out the prevailing model of shallow slab subduction in the middle Cretaceous. The writing is very clear and the figures are very good.

Thank you, Basil.

I feel that this manuscript is, in some ways, very different from the one that I originally reviewed. The authors have cleared up most of the major problems in the original version (e.g., Shatsky conjugate, dealing with the Insular superterrane southerly position, dealing with collisional models, etc.). In their revision, however, they have introduced a few new problems that are not necessary for the main point of this paper. The biggest problem is the last one in this list.

We have addressed the last point by removing the paragraph about sinistral reverse deformation and other references to sinistral deformation in the manuscript. We agree that it is a tighter manuscript without the sinistral deformation component and we will save it for another manuscript.

1. The authors did deal with the “local subduction” problem, but basically incorporating all of northern Mexico in the subduction system. I think you are stretching your claims here. The Peninsular Range magmatic arc is widely acknowledged to turn off at ~90 Ma. There is younger magmatism, but it is not clear how it relates to subduction. In short, by dealing with the Mexican subduction and magmatic history, you are making some interpretations that may not be correct. I read that literature,

but I don't work there and it is hard for me to evaluate the competing models. The point for this manuscript is that it was on firmer ground when they were talking about the Southern California batholith. Please consider paring it down to what you have data about. Alternatively stated, the focus of the original manuscript was better on this issue.

We have reduced some of the parts about Mexican subduction but some remain to show that magmatism is not local in the SCB but is part of a larger magmatic arc system in the Late Cretaceous that continues through northwestern Mexico. This information was included as response to the original review comments from reviewer #3.

2. I am surprised – and gratified – that you took on the hit-and-run model (which was surprisingly more strongly raised by another reviewer). But, because that was not in the original manuscript, it needs some revision.

a. The western Idaho shear zone is off at 85 Ma (at absolute latest). I pointed out this error in my scanned version of the text. In fact, none of the dextral transpressional shear zones in the arcs are active after ~85 Ma. The fault/shear zones that moves the Insular superterrane northwards are – I think – coastward from the western Idaho shear zone and the Sierra Nevada batholith.

Good point. Our writing was unclear and we did not intend to imply that it was active after 85 Ma. We have removed the sentences as suggested in your scanned text edits.

b. I remember from Paul Umhoefer that most people put the Kula-Farallon triple junction at the southern tip of the Insular superterrane. To me, that is the best plate model, because the placement of the North America-Farallon-Kula triple junction is effectively unconstrained and it promotes the northward motion of the Insular superterrane. Your Figure 4 needs to reflect that position (you've adapted it from Yonkee and Weil, who don't really deal with the Insular superterrane). For the record, I think there is yet another plate – the Resurrection – in the mix between the Kula and Farallon plate, but you probably don't have to deal with it.

Another good point. We've modified figure 5 to show the revised location of the Kula-Farallon ridge based on the scanned version.

c. Your model for continued subduction in southern California is, in fact, at odds with the paleomagnetic data from Canada, NW Washington, and Alaska. The paleomagnetism has the southernmost part of the Insular superterrane half way down into Mexico at 100 Ma. That is fine that you are not fully compatible – you can have a different model. But, for scientific fairness, I think that you must acknowledge that your model is not consistent with it (and, for the record, my money is on the paleomagnetism being correct).

We now acknowledge that the paleomagnetic data differs from our model. This is a fair point and we make a point of it in our revised text.

d. You unfairly criticize the hit-and-run model for not being able to deal with a subduction complex (represented by the Pelona-Orocopia schists). In fact, it is exactly compatible – as is the paleomagnetic data – with that interpretation. The Insular terrane must have moved past southern California by 75 Ma on its way northward (it is in place by 55 Ma). That scenario is clearly laid out in Tikoff et al. (2023) and is consistent with Sauer et al. (2019) correlation of the Swakane gneiss of Washington State with the southern California schists.

Our wording did not mean to criticize the hit-and-run model but to provide a possible location for the hit. We agree that it is possible that some schists were translated (e.g., Swakane gneiss of Sauer et al. 2019), although the DZ data do not uniquely fingerprint Southern California as a source (see Jacobson et al. abstract at the 2023 Cordillera GSA). Our point is that the present position of the Pelona, Orocopia and Rand schists beneath the Southern California arc and their moderate P and T metamorphic assemblages indicates that they were underthrust beneath the Southern California batholith. The latter data support renewed subduction after 70 Ma, which is the only point we want to make. We have therefore revised the text so as not to read as a criticism of the hit and run model, but simply to state that our data support continued subduction.

3. The single biggest problem with the manuscript is the inclusion of the sinistral-reverse shear zones present in the southern California batholith. These were absent from the original manuscript, and they should have stayed out of this version. There are no scientific papers that have documented those shear zones that have made it through peer review (that I know of or that are cited). I don't doubt that this team can determine kinematics of shear zones. But, that is not how science works. You can only use data if it is already published or is presented in the manuscript. Neither is applicable. I see exactly why the authors want to utilize it: It appears to make a better story. But, it is inappropriate for a scientific manuscript to utilize unpublished data and I think that not presenting it is going to stop you from making a mistake.

We have removed the main paragraph about sinistral-shear from the manuscript, because as you say, it is not needed for this manuscript. We've also revised to the manuscript and removed reference to sinistral shearing. This addresses the three points below and we agree that the shear zone story requires more development.

a. There is a massive amount of vertical axis rotation in this area of California during the Cenozoic (and probably the Late Cretaceous). While you use the reconstruction of Powell (1991), you don't talk about the reconstruction of those shear zones into a Late Cretaceous position. Their orientation is critical. There is suspected sinistral deformation associated with dextral transpressional "hit" in Idaho, because it is not an orthogonal collision. Rather, the orientation of the boundary is going to control whether a shear zone is sinistral or dextral. If there is any place in the Cordillera where a sinistral-reverse zone will happen during the initial obliquely dextral collision of the Insular block, it is going to be in southern California.

As discussed above, we agree and will save the discussion of reconstruction of shear zones for another manuscript.

b. There is no documentation of the timing of the shear zones, and the timing information is essential. You make a point that you think that the reverse-sinistral shear zones might record the initiation of the shallow slab subduction associated with Hess conjugate (I think that intuition is likely correct). But, it all hinges on their timing, which you DO NOT constrain. If they are ~75 Ma, they are almost certainly the result of trailing edge subduction.

We were not clear and did not emphasize enough in prior versions that titanite dates from the shear zones shown in prior version of Supplementary Figure 5 were from syn-kinematic titanites in the Black Belt and Tumamait shear zones. The titanites give ages of 83 Ma and 77-74 Ma respectively, which supports syn-magmatic, intra-arc deformation. Nonetheless, we will save these dates for another manuscript and we have removed them from this version.

c. Sinistral shear zones can accommodate dextral motion. One example is antithetic block rotation. A second example, as argued in Tikoff et al. (2023b), the sinistral motion on the Lewis and Clark line is a result of the larger dextral motion. That sinistral motion starts at ~85 Ma, which may be consistent with your timing.

This is a good point and the tectonic implications of sinistral deformation need to be explored in more detail. Again, we have removed the discussion of the sinistral-reverse shear zones from this manuscript.

The most important reason to not present information about the shear zones is that you don't need to. Your zircon data stands on its own as a completely compelling case for there not being shallow slab subduction until after 75 Ma (or never, in my opinion). You have great structural geologists on your team. Let them publish the sinistral-reverse story when they know: 1) What was the orientation of the shear zones when they were active; 2) When were they active; and 3) How it relates to the larger tectonic context. Besides, it gives your team a second major paper to write. You are – in my opinion – making an unforced error by trying to deal with the shear zones in this paper.

We agree about removing the sinistral-reverse shear zone discussion from this manuscript and we will address it in another manuscript.

I completely accept your main point: There cannot be a shallow slab in southern California until after 75 Ma. Your data support that. I think your model is incorrect for continued subduction until 75 Ma in southern California, but I must admit that it is admissible with your data. Anything after 75 Ma is conjecture, and is not the strength of this manuscript.